# Extracellular calcium functions as a molecular glue for transmembrane helices to activate the scramblase Xkr4

Panpan Zhang [1,2], Masahiro Maruoka [1,3], Ryo Suzuki [4], Hikaru Katani [1], Yu Dou[1,2], Daniel M. Packwood [1], Hidetaka Kosako [5], Motomu Tanaka [1,4,6] & Jun Suzuki [1,2,3,7]

The "eat me" signal, phosphatidylserine is exposed on the surface of dying cells by phospholipid scrambling. Previously, we showed that the Xkr family protein Xkr4 is activated by caspase-mediated cleavage and binding of the XRCC4 fragment. Here, we show that extracellular calcium is an additional factor needed to activate Xkr4. The constitutively active mutant of Xkr4 is found to induce phospholipid scrambling in an extracellular, but not intra-cellular, calcium-dependent manner. Importantly, other Xkr family members also require extracellular calcium for activation. Alanine scanning shows that D123 and D127 of TM1 and E310 of TM3 coordinate calcium binding. Moreover, lysine scanning demonstrates that the E310K mutation-mediated salt bridge between TM1 and TM3 bypasses the requirement of calcium. Cysteine scanning proves that disulfide bond formation between TM1 and TM3 also activates phospholipid scrambling without calcium. Collectively, this study shows that extracellular calcium functions as a molecular glue for TM1 and TM3 of Xkr proteins for activation, thus demonstrating a regulatory mechanism for multi-transmembrane region-containing proteins.

Phospholipids are asymmetrically distributed in the lipid bilayer of plasma membranes, wherein phosphatidylserine (PS) and phosphatidylethanolamine (PE) locate at the inner leaflet of the membranes while phosphatidylcholine (PC) and sphingomyelin (SM) locate at the outer leaflet[1,2]. To maintain the asymmetrical distribution of phospholipids, P4-ATPase functions as a flippase to translocate PS and PE from the outer to the inner leaflet in an ATP-dependent manner[3]. However, asymmetrical distribution of phospholipids is altered in various physiological situations to adapt to the environmental changes. For example, PS is exposed on the cell surface of activated platelets by phospholipid scrambling (PLS) and functions as a scaffold for

coagulation factors[4]. PS is also exposed on the apoptotic cell surface, where it functions as an "eat me" signal[5,6] for apoptotic cells to be engulfed by phagocytes. In these processes, scramblases non-specifically and bi-directionally translocate phospholipids in the lipid layer without energy consumption[7–9].

We previously identified two families of scramblases: the TMEM16 family[10,11], calcium-dependent scramblases, and the XK-related (Xkr) family[12,13], caspase cleavage-dependent scramblases. The Xkr family proteins are comprised of ten transmembrane (TM) helices including two α-helices which are partly embedded in the membrane: one is positioned between TM2 and TM3, and another between TM6 and

[1]Institute for Integrated Cell-Material Sciences (WPI-iCeMS), Kyoto University, Yoshida-Honmachi, Sakyoku, Kyoto 606-8501, Japan. [2]Graduate School of Biostudies, Kyoto University, Konoe-cho, Yoshida, Sakyoku, Kyoto 606-8501, Japan. [3]Center for Integrated Biosystems, Institute for Biomedical Sciences, Academia Sinica, Taipei, Taiwan. [4]Center for Integrative Medicine and Physics (CiMPhy), Institute for Advanced Study, Kyoto University, Yoshida-Honmachi, Sakyoku, Kyoto 606-8501, Japan. [5]Fujii Memorial Institute of Medical Sciences, Institute of Advanced Medical Sciences, Tokushima University, 3-18-15 Kuramoto-cho, Tokushima 770-8503, Japan. [6]Physical Chemistry of Biosystems, Institute of Physical Chemistry, Heidelberg University, 69120 Heidelberg, Germany. [7]CREST, Japan Science and Technology Agency, Kawaguchi, Saitama 332-0012, Japan. ✉e-mail: jsuzuki@icems.kyoto-u.ac.jp

TM7[14,15]. To initiate PLS activity, Xkr4, Xkr8 and Xkr9 are cleaved by caspases at a conserved caspase-cleavage site during apoptosis[13]. Among these members, Xkr8 forms a heterodimer with Basigin (BSG) or Neuroplastin (NPTN) and forms a heterotetramer after caspase-mediated cleavage[16]. On the other hand, Xkr4 stays as a monomer in its resting state, and forms a homodimer after caspase-mediated cleavage[17]. In the case of Xkr4, dimerization is not sufficient for Xkr4 to be activated: the caspase-cleaved fragment of XRCC4 is required for Xkr4 to be activated[17]. However, the combination of Xkr4 dimerization with the XRCC4 fragment binding failed to completely activate Xkr4, suggesting another factor is necessary to fully activate Xkr4.

Here, we identified extracellular $Ca^{2+}$ as the additional factor needed to activate Xkr4. Both Xkr4 monomer treated with apoptotic stimuli and Xkr4 dimer stimulated with XRCC4 introduction required extracellular calcium to induce PLS. The constitutively-active mutant of Xkr4 also induced PLS in an extracellular calcium-dependent manner. Similarly, Xkr8 and Xkr9 required extracellular $Ca^{2+}$ for their activation. Unexpectedly, Ala scanning showed that a $Ca^{2+}$ binding site of Xkr4 locates within TM regions. Cys scanning to introduce a disulfide bond and Lys scanning to introduce a salt bridge between TM1 and TM3 showed that PLS is induced even without $Ca^{2+}$ when TM1 and TM3 contact is artificially generated. These results demonstrated that $Ca^{2+}$ functions as a molecular glue to connect TM1 and TM3 for activation of Xkr family members.

## Results

### Requirement of extracellular calcium for Xkr4-mediated PLS

Xkr4 forms a dimer by caspase-mediated cleavage of the C-terminus and is then activated by binding of the caspase-cleaved XRCC4 fragment during cell death[13,17]. We induced cell death with staurosporine (STS) in PLB cells expressing either the full-length (FL) or the caspase-cleaved form ($\Delta$C) of Xkr4 and analyzed phospholipid scrambling (PLS) activity by assaying incorporation of extracellularly-added fluorescent PC (NBD-PC). Unexpectedly, PLS activity was lost when the extracellular divalent cations were removed in Xkr4FL and Xkr4$\Delta$C-expressing cells, but addition of $Ca^{2+}$ rescued the phenotype (Fig. 1a), suggesting that $Ca^{2+}$ is required during the Xkr4 activation process.

In our previous study, we found that the caspase-cleaved fragment of XRCC4 binds to Xkr4 directly to induce PLS activity after apoptotic stimulation[17]. To investigate whether this caspase activation is essential for $Ca^{2+}$ to induce PLS activity, we established the Xkr4 activation system in living cells using the XRCC4 C-terminal fragment (XRCC4/C) purified from *E.coli*. XRCC4/C was introduced into Ba/F3 cells deficient in both *Xkr8* and *TMEM16F* (BDKO) expressing Xkr4FL or Xkr4$\Delta$C using electroporation. PLS activity was induced in Xkr4$\Delta$C-expressing cells when XRCC4/C was introduced in the presence of $Ca^{2+}$ and this activity was maintained for at least for 6 h (Fig. 1b and Supplementary Fig. 1d). In our previous study, the R270A or I266G mutants of XRCC4 failed to activate Xkr4[17]. Consistent with this, the purified XRCC4/C with mutations R270A or I266G hardly induced PLS activity when introduced with electroporation (Supplementary Fig. 1a-d). These results demonstrate that $Ca^{2+}$ is required for Xkr4 activation downstream of caspase activation.

To investigate the effect of calcium further, the constitutively active mutant Xkr4$\Delta$C Q332E (aXkr4)[17], which does not require XRCC4/C for activation, was applied to the PLS analysis. Similarly, $Ca^{2+}$ was required for Xkr4-mediated PLS of PC as well as SM and PS (Fig. 1c and Supplementary Fig. 1e, f) in an extracellular $Ca^{2+}$ concentration-dependent manner (Fig. 1d), suggesting that $Ca^{2+}$ is directly involved in Xkr4 activation. In the presence of 125 μM of extracellular calcium, aXkr4 activity is increased but was not enhanced by treatment with the calcium ionophore A23187 (Fig. 1e), suggesting that extracellular, but not intracellular, calcium is required to activate Xkr4. Consistent with this, treatment of aXkr4-expressing cells with thapsigargin, used to increase intracellular $Ca^{2+}$, failed to induce PLS

(Supplementary Fig. 1g, h), confirming the significance of extracellular calcium to activate Xkr4. To examine the effect of extracellular $Ca^{2+}$-mediated PLS on engulfment, Xkr4-expressing cells were treated with ultraviolet (UV) to induce apoptosis, labeled with pHrodo, and incubated with macrophages. Flow cytometry analysis showed that Xkr4-expressing apoptotic cells were efficiently engulfed by macrophages in the presence of $Ca^{2+}$, but not in the absence of it (Supplementary Fig. 1i), indicating that extracellular $Ca^{2+}$ functions as a molecular switch for cell clearance via PLS.

PLS-mediated changes in lipid distribution alter membrane tension[18]. To examine the effect of PLS on membrane tension, we measured the tension of cells expressing aXkr4 with an optical trap in the presence or absence of $Ca^{2+}$ (Fig. 1f). Indeed, $Ca^{2+}$-mediated PLS in aXkr4-expressing cells reduced membrane tension (Fig. 1g, h), further supporting that extracellular $Ca^{2+}$ is required for Xkr4-mediated PLS (Fig. 1i).

### Selectivity of metal cations on Xkr4 activation

Calcium-binding sites of proteins often function to bind other divalent and trivalent cations which have similar ionic radii or properties to $Ca^{2+}$ such as $Sr^{2+}$, $Mn^{2+}$, and $Tb^{3+}$ [19]. To gain insight into the selectivity of metal ions, we examined other ions which may allow Xkr4 activation. BDKO cells expressing aXkr4 were incubated with 1 mM of different cations and their PLS activity was examined. Among several divalent cations, $Sr^{2+}$ induced the 2nd strongest PLS activity after $Ca^{2+}$ (Fig. 2a). $Sr^{2+}$ possessed higher activation capability than other ions, but still showed about a 7-fold decrease in affinity compared to $Ca^{2+}$ (Fig. 2b, c). $Mn^{2+}$ and $Tb^{3+}$ weakly activated aXkr4, but showed a drastic decrease in affinity compared to $Ca^{2+}$ (Fig. 2b, c). In contrast, $Mg^{2+}$ and $Ba^{2+}$ failed to induce PLS. Considering the ionic radius of cations, $Ba^{2+}$ (0.136 nm), $Sr^{2+}$ (0.125 nm), $Ca^{2+}$ (0.100 nm), $Tb^{3+}$ (0.092 nm), $Mn^{2+}$ (0.083 nm), $Mg^{2+}$ (0.072 nm)[20–22], appropriate ionic radius (0.083−0.125 nm) is required for cations to bind to the calcium-binding site. These results suggest that the calcium-binding site is flexible to connect TM1 and TM3 in the presence of cations, but $Ca^{2+}$ is the best molecule to connect them to activate Xkr4.

### Potential calcium-binding sites

Among Xkr family members, Xkr8 and Xkr9 also induce PLS via caspase-mediated cleavage of their C-terminal region[12,13]. When mouse and human Xkr8- or Xkr9-expressing PLB cells were treated with UV to induce apoptosis, PLS was observed in an extracellular $Ca^{2+}$- or $Sr^{2+}$-dependent manner, similar to what was observed for Xkr4 (Fig. 3a and Supplementary Fig. 2b). These results suggest that the calcium requirement for PLS is conserved in these Xkr family members in both humans and mice.

To search for potential calcium-binding sites, we aligned mouse and human Xkr4, Xkr8, and Xkr9, focusing on conserved negatively-charged amino acids (Fig. 3b and Supplementary Fig. 2a). Generally, extracellular calcium is believed to bind to the extracellular regions of proteins while intracellular calcium binds to the cytoplasmic regions. Although we searched for calcium-binding sites in the extracellular region, candidate sites were not determined. In the case of the multi-transmembrane (TM)-containing protein TMEM16 family, requiring intracellular calcium, calcium-binding sites locate on TM regions[23,24]. Considering this, we speculated that calcium-binding sites of Xkr family members may exist on TM regions accessible from the extracellular space. We then marked the negatively charged amino acids on TM regions to select candidate amino acid residues. While considering the 3D structure of Xkr8 and Xkr9[14,15] and a predicted structure of mouse Xkr4 by the RoseTTAFold server[25], validated by the software verify 3D and PROCHECK, potential calcium-binding sites were identified: D123 and D127 on TM1 and E310 on TM3 of Xkr4 coordinate a pocket potentially accessible from the extracellular region (Fig. 3c, d, and Supplementary Fig. 2c, d). Supporting this hypothesis, side chains

 

of the three amino acids were positioned towards the inside of TM regions, suggesting that D123, D127, and E310 form a potential calcium-binding site.

## Testing the candidate calcium-binding sites

After narrowing down the candidate calcium-binding sites, we investigated the effect of these residues on Xkr4 activity. The candidate amino acids were mutated in aXkr4, and the mutants were expressed in BDKO cells to examine PLS activity. At first, to avoid Xkr4 instability caused by the drastic change in side chains, we neutralized the amino acid to examine the effects of the negative charge for three amino acids. Among these, two mutants, D127N and E310Q, significantly reduced the activity of Xkr4 (Fig. 4a, b) with reduced affinity toward $Ca^{2+}$, indicating that the charged oxygen participates in forming a calcium-binding site in D127 and E310. Compared to these two, mutant

D123N showed a small decrease in PLS activity (Fig. 4a, b), suggesting that the non-charged oxygen of D123 forms a calcium-binding site. To understand the effects of the side chain further, we generated Ala mutants of each acidic residue. As a result, all three mutants (D123A, D127A, and E310A) lost PLS activity and showed low affinity to $Ca^{2+}$ (Fig. 4a, c). These mutants also showed a similar tendency for $Sr^{2+}$-mediated PLS (Fig. 4d, e), suggesting the critical role of D123, D127, and E310 for coordinating a $Ca^{2+}$-binding site. To examine the effect of these amino acids, the E310A mutant was selected for further analysis. The membrane tension of cells expressing aXkr4 E310A was examined with an optical trap in the presence or absence of $Ca^{2+}$. The results showed that $Ca^{2+}$-mediated PLS in aXkr4 E310A-expressing cells was not altered even with 5 mM $Ca^{2+}$, further supporting that the extracellular $Ca^{2+}$ is required for Xkr4-mediated PLS through D123, D127, and E310 (Fig. 4f, g).

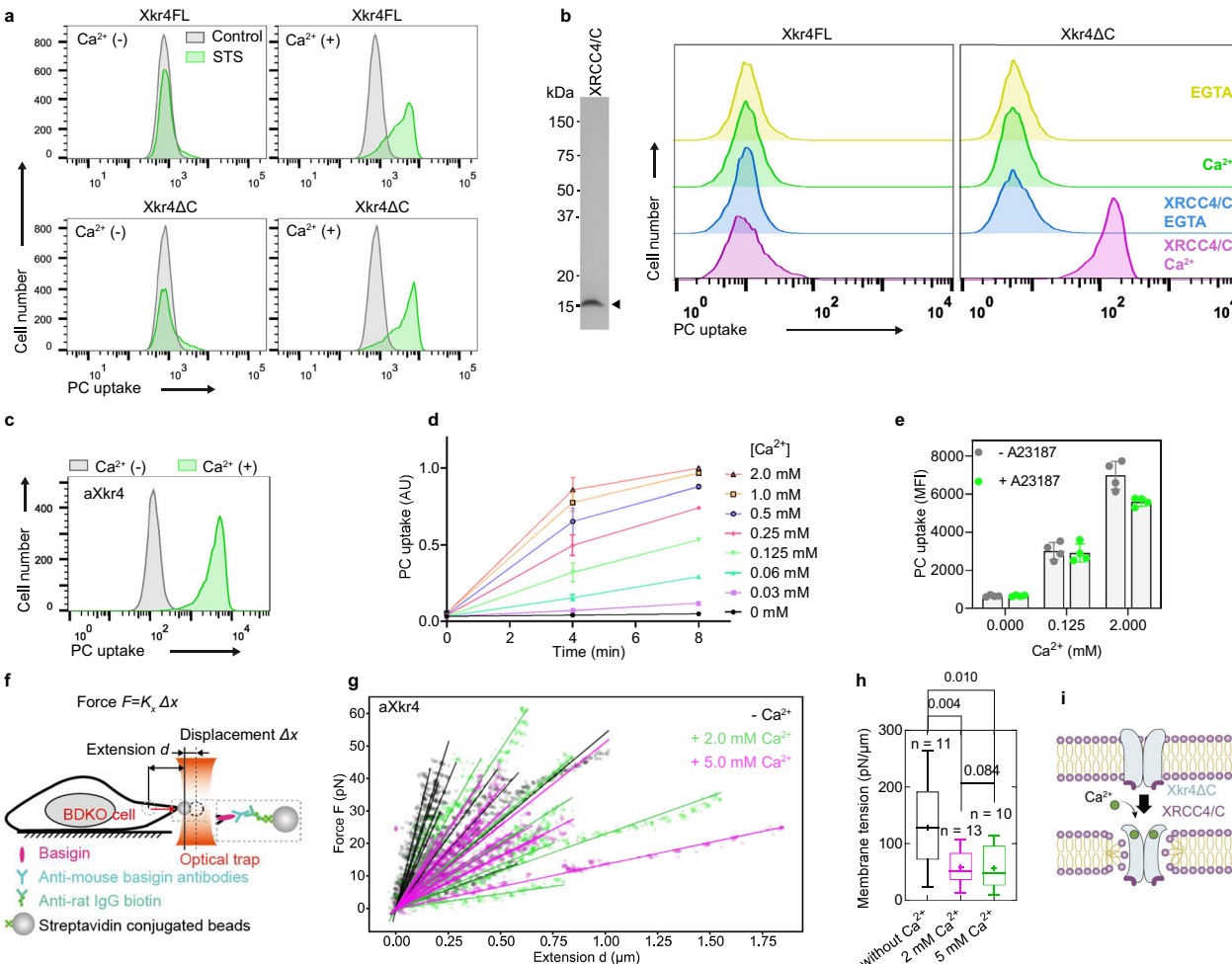

**Fig. 1 | Requirement of extracellular calcium for Xkr4-mediated PLS. a** PC uptake assay of PLB cells expressing Xkr4FL or Xkr4ΔC. Cells were stimulated with 10 μM STS to induce apoptosis, incubated with NBD-PC in the presence or absence of 1 mM $Ca^{2+}$, and analyzed by FACS. **b** PC uptake assay of BDKO cells expressing Xkr4FL or Xkr4ΔC. Cells were electroporated with 0.5 μM XRCC4/C, cultured for 1 h, incubated with NBD-PC in the presence of 1 mM $Ca^{2+}$ or 0.5 mM EGTA, and analyzed by FACS (right). XRCC4/C purified from bacteria was applied to SDS-PAGE and CBB staining (left). **c** PC uptake assay of BDKO cells expressing aXkr4. Cells were incubated with NBD-PC in the presence or absence of 1 mM $Ca^{2+}$ for 10 min. **d** Effect of $Ca^{2+}$ concentration on Xkr4 activity. BDKO cells expressing aXkr4 were incubated with NBD-PC in the presence of various $Ca^{2+}$ concentrations for 4 or 8 min and analyzed by FACS. Averages of triplicates were shown with SD. AU represents arbitrary units. **e** Effect of intracellular $Ca^{2+}$. BDKO cells expressing aXkr4 were treated with the calcium ionophore A23187 at 1 μM in the presence of various

$Ca^{2+}$ concentrations for 4 min and analyzed by FACS. Averages of quadruplicates were shown with SD. MFI, mean fluorescence intensity. **f** Schematic model of membrane tension measurement with optical trap. **g** Force-extension curve used to obtain membrane tension for BDKO cells expressing aXkr4 in the presence of 2 mM (pink) or 5 mM $Ca^{2+}$ (green), or absence of it (black). Solid lines are the linear fit used to obtain membrane tension. **h** Membrane tension of BDKO cells expressing aXkr4. Box-plots of membrane tension aXkr4 in the presence of 2 mM (pink) or 5 mM $Ca^{2+}$ (green), or absence of it (black). Statistical test was performed using a two-tailed Student's *t* test. p < 0.05 was considered statistically significant. In the box plots, the center line indicates the median, the plus character (+) in the center indicates the mean value, the edges of the box indicate the first and third quartiles, the upper whisker indicates the maxima value, and lower whisker indicates the minima value. **i** A model of Xkr4 activation. Extracellular $Ca^{2+}$ binds to the Xkr4 dimer interacted with XRCC4/C. Source data are provided as Source Data file.

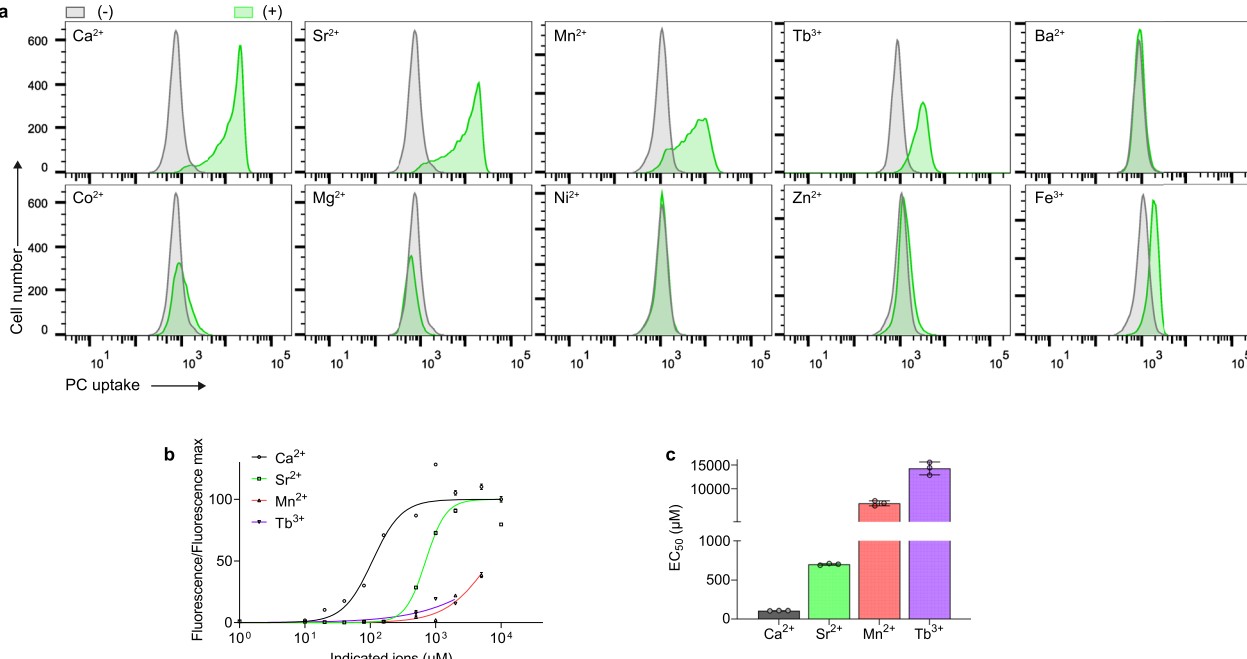

**Fig. 2 | Selectivity of metal cations on Xkr4 activation. a** Effect of different cations on Xkr4 activity. BDKO cells expressing aXkr4 were incubated with NBD-PC in the presence of 1 mM $CaCl_2$, $SrCl_2$, $MnCl_2$, $TbCl_3$, $BaCl_2$, $FeCl_3$, $CoCl_2$, $MgCl_2$, $NiCl_2$, or $ZnCl_2$ and analyzed by FACS. **b** Dose-response curves of Xkr4 activity based on PC uptake. BDKO cells expressing aXkr4 were incubated with NBD-PC in the presence of a series of concentrations of $CaCl_2$, $SrCl_2$, $MnCl_2$, or $TbCl_3$ and analyzed by FACS, and quantified. Averages of triplicates were shown with SD. The data for $Mn^{2+}$ at 10 mM and $Tb^{3+}$ at 5 mM and 10 mM were not used for analysis because cells died. **c** Half maximal effective concentration (EC50) of indecated cations on Xkr4 activity. EC50 was calculated from the dose-response curve of (**b**) and shown for each cation. Averages of triplicates were shown with SD. Source data are provided as Source Data file.

$Ca^{2+}$ generally interacts and is stabilized with 6 to 8 oxygen atoms in proteins. To examine whether other amino acid residues in Xkr4 coordinate a $Ca^{2+}$-binding site, we investigated the modeled Xkr4 structure focusing on the region around D123, D127, and E310, and found that T307 and S311 in TM3, and S339 in TM4 (Supplementary Fig. 3a) faced towards the $Ca^{2+}$-binding site. When these residues were substituted to Ala, all mutants failed to induce the PLS activity (Supplementary Fig. 3b). Among these, T307 and S339, highly conserved in Xkr8 and Xkr9, decreased the affinity to $Ca^{2+}$ more significantly than S311 after Ala substitution (Supplementary Fig. 3c, d), suggesting that T307 and S311 on TM3, and S339 on TM4 support D123 and D127 on TM1 and E310 on TM3 to form a calcium-binding site.

### Interaction between transmembrane 1 and 3
In the case of the TMEM16 family, two $Ca^{2+}$ accessed from the intracellular space connect TM6, TM7, and TM8 and one $Ca^{2+}$ connects TM2 and TM10 for activation[26–28]. We speculated that similarly, $Ca^{2+}$ functions as a "molecular glue" to connect TM1 and TM3 to activate Xkr proteins. To examine this hypothesis, we tried to connect TM1 and TM3 by an induced-disulfide bond through Cys-scanning analysis (Fig. 5a). In the context of the E310C mutation in TM3 of aXkr4, amino acid residues from D123 to D127 of TM1 were replaced with Cys sequentially (Fig. 5b). Single Cys mutants and double Cys mutants without oxidative reagents were not sufficient to activate aXkr4, suggesting that a disulfide bond is not spontaneously formed (Supplementary Fig. 4a, b). To induce Cys residues to form a disulfide bond, we treated the cells with the oxidative reagent copper-phenanthroline (CuP)[29]. Although single Cys mutants in TM1 without E310C of TM3 failed to activate aXkr4 with the CuP treatment (Fig. 5c), some mutants with E310C showed strong PLS activity. Especially, the double Cys mutant G125C/E310C induced obvious PLS activity after CuP treatment (Fig. 5d, e), suggesting that the Cys residues in the double mutant G125C/E310C form a disulfide bond. Similarly, the G125C/E310C

mutant showed PLS activity after treatment with iodine solution to induce disulfide bond formation[30] (Fig. 5f–h). These results demonstrated that aXkr4 was activated by disulfide bond formation between TM1 and TM3. To confirm that the connection between TM1 and TM3 induces PLS, Arg and Lys mutations were introduced into aXkr4 to facilitate the salt bridge formation in the $Ca^{2+}$-binding site. Among the mutants, E310K induced significant PLS activity without $Ca^{2+}$ and the E310R had a weak effect on PLS activity (Fig. 5i). Together, these results substantiate our hypothesis that $Ca^{2+}$ functions as a molecular glue to connect D123 and D127 of TM1 to E310 of TM3.

$Ca^{2+}$ binding may cause conformational changes of Xkr4 through TM1 and TM3 arrangement. Thus, we next checked whether $Ca^{2+}$ binding to Xkr4 affects the stability of the Xkr4 protein. EGFP-fused Xkr4ΔC and aXkr4 were expressed in HEK293T cells and solubilized in a mixture of GDN and LMNG at a ratio of 1:1, and dimer fraction was collected by size exclusion chromatography. The collected proteins were then applied to thermostability analysis at different temperatures in the presence of $Ca^{2+}$ or EGTA. As a result, aXkr4, but not Xkr4ΔC, showed less thermostability without $Ca^{2+}$: at an elevated temperature, the Xkr4 dimer was dissociated into a monomer (Supplementary Fig. 5a–d). One interpretation for this result is that $Ca^{2+}$ binding occurs at the aXkr4, but not at the Xkr4 dimer, during the activation process.

### Activation mechanisms of Xkr4
Next, we tried to examine whether the contact formation between TM1 and TM3 is sufficient to activate the PLS activity of Xkr4. The Cys double mutations G125C/E310C were inserted into Xkr4FL and Xkr4ΔC, and an artificial disulfide bond was induced by CuP or $I_2$ treatment. The results showed that Xkr4FL failed to induce PLS activity. Although Xkr4ΔC slightly induced PLS activity after CuP or $I_2$ treatment, the activity was much less than that of aXkr4 G125C/E310C (Fig. 6a–c), suggesting that binding of XRCC4/C to Xkr4ΔC is critical for $Ca^{2+}$-mediated activation of the Xkr4 dimer. Indeed, introduction of

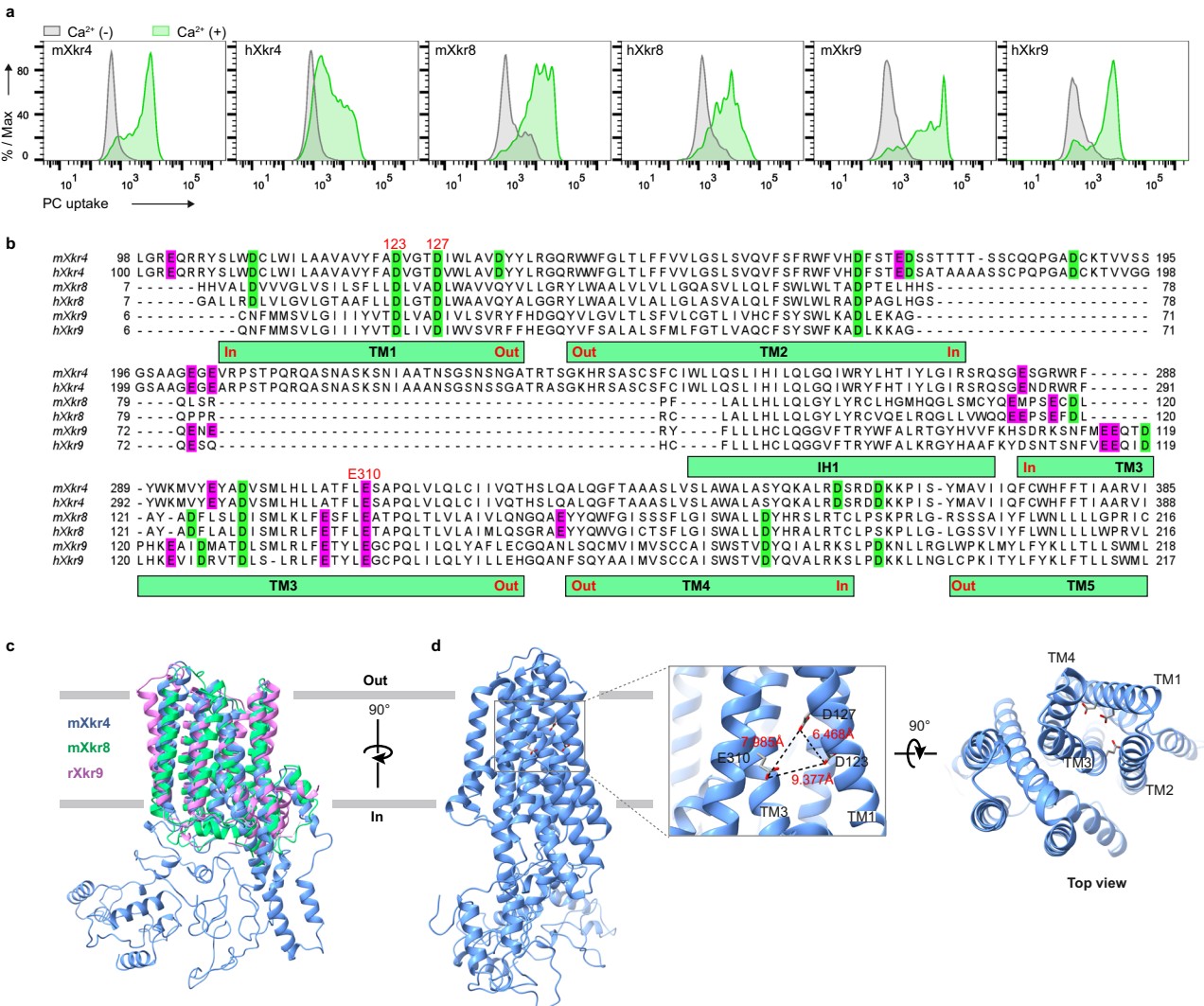

**Fig. 3 | Potential calcium-binding sites. a** PC uptake assay of PLB cells expressing mouse (m) or human (h) Xkr4, Xkr8, or Xkr9. Cells were stimulated with UV (Xkr4: 2000 J/m², Xkr8 and Xkr9: 200 J/m²), cultured for 3 h (Xkr4) and 2 h (Xkr8 and Xkr9), and applied to the NBD-PC assay in the presence or absence of 1 mM Ca²⁺. **b** Sequence alignment of mouse and human Xkr4, Xkr8, and Xkr9. Highly conserved acidic residues among Xkr4, Xkr8, and Xkr9 are highlighted in green for Asp and purple for Glu. The potential calcium-binding residues were labeled with a red character numbered in mXkr4 position. **c** Structural comparison of mouse Xkr8 (springgreen, PDB: 7DCE), rat Xkr9 (violet, PDB:7P16), and mXkr4 predicted by RoseTTAFold, where the cytoplasmic loop was modified by Modeler. Viewed parallel to the membrane. **d** A calcium-binding site of mXkr4. The potential calcium-binding residues were shown as sticks (oxygen: red), and the distances of each residue were represented by dash lines.

XRCC4/C into Xkr4ΔC G125C/E310C-expressing cells activates the Xkr4 dimer after CuP or I₂ treatment (Fig. 6d and Supplementary Fig. 6a). When the E310K mutation was inserted into Xkr4FL and Xkr4ΔC, their activity was also much less than of aXkr4 E310K (Supplementary Fig. 6b) From these results, we postulate a new model for Xkr4 activation: 1. Caspase-mediated cleavage of Xkr4 induces dimer formation; 2. The caspase-cleaved fragment of XRCC4 binds to the Xkr4 dimer to prepare for activation; 3. Extracellular Ca²⁺ coordinates the contact formation between TM1 and TM3 for activation of Xkr4 (Fig. 6e, f).

## Discussion

Xkr4, Xkr8, and Xkr9 among Xkr family members are scramblases activated by caspase-mediated cleavage of their C-terminus. Xkr8 forms a heterodimer with BSG or NPTN under resting condition and forms a hetero-tetramer after caspase-mediated cleavage of the C-terminus[16]. In contrast, Xkr4 stays as a monomer and forms a homodimer after caspase-mediated cleavage. Although Xkr4

dimerization alone is not enough for activation, binding of the C-terminal fragment of XRCC4 to the Xkr4 dimer activates PLS activity[17]. As for Xkr8 and Xkr9, they are quickly activated without the XRCC4 fragment during cell death[13,17]. However, all Xkr4, Xkr8, and Xkr9 were found to require extracellular Ca²⁺ for their activation. Through Ala-scanning and Cys-scanning, we found that three negatively charged residues, D123 and D127 in TM1 and E310 in TM3, of Xkr4 coordinate a Ca²⁺-binding site with T307 and S311 in TM3, and S339 in TM4. Most of these amino acids are conserved in human and mouse Xkr4, Xkr8, and Xkr9, suggesting a critical role for Ca²⁺-mediated Xkr activation.

TMEM16 family members are Ca²⁺-dependent proteins, many of which function as scramblases[10,11]. In the TMEM16 family, two Ca²⁺-binding sites were found in one TMEM16 molecule, one of which binds in TM 6, 7, and 8, and another in TM2 and 10, which leads to a conformation change that induces PLS activity. In particular, TM4 and 6 form a hydrophilic groove for translocation of the phospholipid head[26,27,31]. Analysis of the activated form of TMEM16 proteins

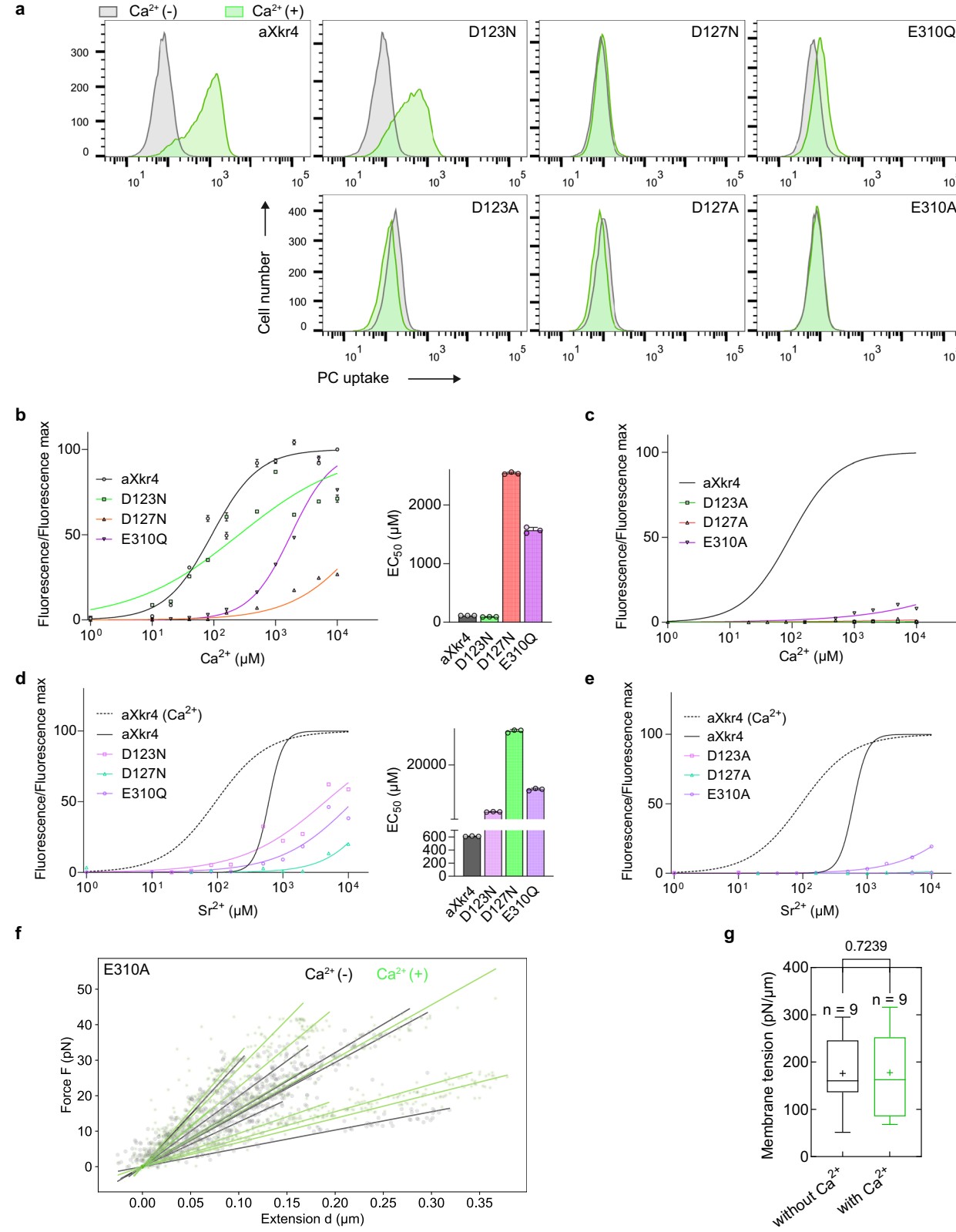

suggested that calcium binding to the protein altered a hydrophilic groove providing a hydrophilic surface for the head group of phospholipids.

Recently, the monomeric structures of Xkr8 and Xkr9 have been revealed[14,15], which share a similar structure to the predicted structure of mouse Xkr4, especially in the transmembrane region (Fig. 3c). Currently, the Nagata and Dutzler groups have suggested the potential

phospholipid translocation pathways in Xkr8 and Xkr9, respectively. Combining the structural and functional analysis of Xkr8 and Xkr9 with this study, we could understand that regulation of TM3 in Xkr proteins may be the key to induce PLS. Indeed, TM3 possesses highly conserved polar and charged residues, three of which (T307, E310, and S311 in Xkr4) is used to coordinate a $Ca^{2+}$-binding site with D123 and D127 in TM1 and S339 in TM4 to initiate PLS. An artificial disulfide bond formation or

**Fig. 4 | Testing the candidate calcium-binding sites. a** PC uptake assay of BDKO cells expressing aXkr4 with single mutations. Cells expressing each mutant (D123N, D127N, E310Q, D123A, D127A, and E310A) were incubated with NBD-PC at a final concentration of 0.125 mM $Ca^{2+}$ or without it and applied to FACS. **b–e** Dose-response curves of Xkr4 activity based on PC uptake. BDKO cells expressing aXkr4 with single mutation D123N, D127N or E310Q (**b, d**) or D123A, D127A, or E310A (**c, e**) were incubated with NBD-PC at various $Ca^{2+}$ (**b, c**) or $Sr^{2+}$ (**d, e**) concentration and analyzed by FACS. EC50 is shown on the right (**b, d**). Averages of triplicates were shown with SD. The results of aXkr4 in (**c**) and (**e**) are the same as (**b**) and (**d**), respectively. aXkr4 for $Ca^{2+}$ in (**d**) and (**e**) is shown as dotted line. **f** Force–extension

curve used to obtain membrane tension for BDKO cells expressing aXkr4 E310A in the presence (green) or absence (black) of 5 mM $Ca^{2+}$. Solid lines are the linear fit used to obtain membrane tension. **g** Membrane tension of BDKO cells expressing aXkr4 E310A. Box plots of membrane tension in the presence (green) or absence (black) of 5 mM $Ca^{2+}$. Statistical test was performed using a two-tailed Student's $t$ test. $p < 0.05$ was considered statistically significant. In the box plots, the center line indicates the median, the plus character (+) in the center indicates the mean value, the edges of the box indicate the first and third quartiles, the upper whisker indicates the maxima value, and lower whisker indicates the minima value. Source data are provided as Source Data file.

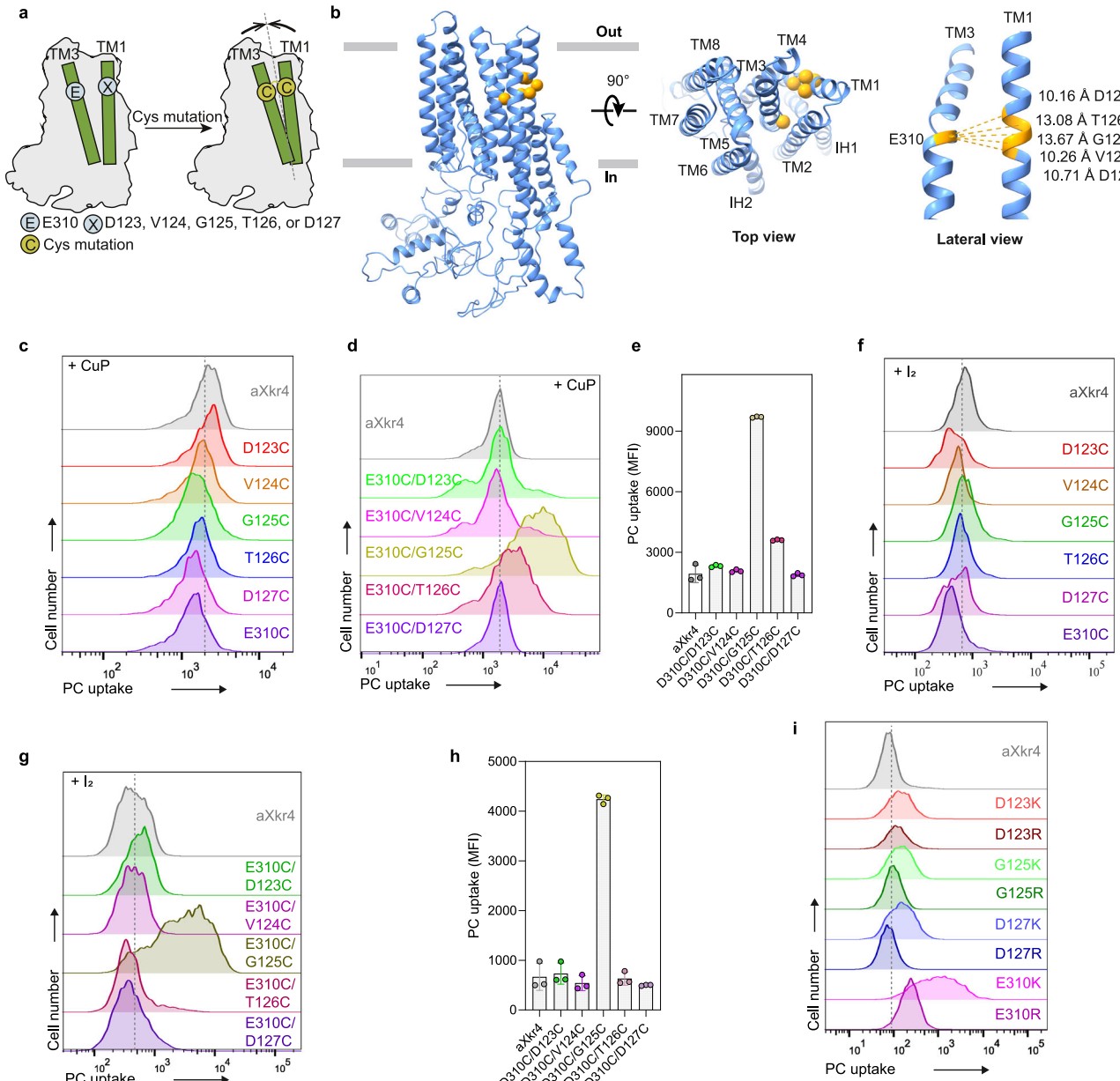

**Fig. 5 | Interaction between transmembrane 1 and 3. a** Schematic representation of Cys-scanning in TM1 and TM3 of Xkr4. The calcium binding sites of Xkr4 were mutated into Cys. **b** A 3D model of predicted mXkr4 of top and lateral view. Cys mutation sites were highlighted as yellow spheres (α carbon). The distances between E310 on TM3 and D123-D127 on TM1 were represented by yellow dash lines. **c–h** Effect of Cys single mutations on Xkr4 activity. BDKO cells expressing aXkr4 with indicated Cys single mutations in combination with E310C mutation

(**d, g**) or without it (**c, f**) were treated with a final concentration of 1.1 mM CuP in TBS buffer at room temperature for 20 min (**c, d**) or with a final concentration 25 µM iodine solution in TBS buffer at room temperature for 10 min (**f, g**). Cells were then applied to the NBD-PC assay without $Ca^{2+}$ and analyzed by FACS. Triplicate data were shown in right with SD (**e, h**). **i** Effect of Arg or Lys mutation on Xkr4 activity. BDKO cells expressing aXkr4 with indicated mutations were applied to the NBD-PC assay without $Ca^{2+}$. Source data are provided as Source Data file.

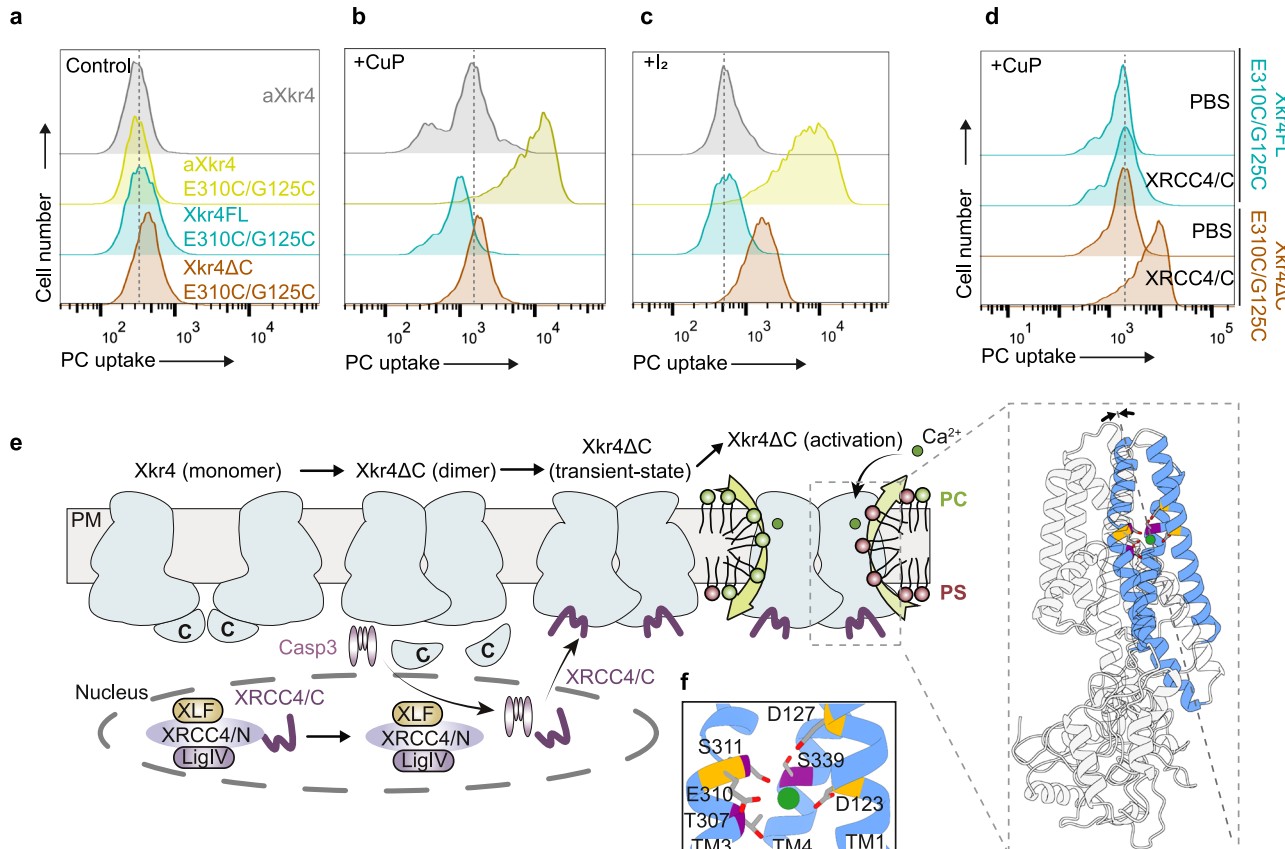

**Fig. 6 | Activation mechanisms of Xkr4. a–d** Effect of Cys double mutations on Xkr4 activity. BDKO cells expressing Xkr4FL, Xkr4ΔC, or aXkr4 with G125C/E310C mutations (**a**) were treated with a final concentration of 1.1 mM CuP in TBS buffer at room temperature for 20 min (**b**) or with a final concentration of 25 μM iodine solution in TBS buffer at room temperature for 10 min (**c**). Cells were then applied to the NBD-PC assay without $Ca^{2+}$ and analyzed by FACS. Cells expressing Xkr4FL or Xkr4ΔC G125C/E310C were electroporated with 0.5 mM XRCC4/C or PBS buffer for control and were treated with a final concentration of 1.1 mM CuP in TBS buffer at room temperature for 20 min (**d**). **e** A model of Xkr4 activation. Xkr4 forms a homodimer after cleavage of the C-terminus by caspase3 (Casp3). XRCC4/C generated by Casp3-mediated cleavage binds to Xkr4 dimer. $Ca^{2+}$ binds to D123 and D127 in TM1, and E310 in TM3 to activate Xkr4. The negatively-charged residues (D123, D127, E310) were highlighted as orange, and the non-charged residues for calcium-binding were highlighted as purple (oxygen: colored in red at the tips of gray sticks). Lime green circle, $Ca^{2+}$. XRCC4/N, N-terminus of XRCC4. LigIV, DNA ligaseIV. **f** Close-up view of $Ca^{2+}$ -binding site.

a salt bridge formation between TM1 and TM3 of aXkr4 induced PLS, suggesting that $Ca^{2+}$ might function as a "molecular glue" to connect TM1 and TM3 (Fig. 5d, g, i). Thus, TM3 might interact with the phospholipid head or generate a hydrophilic surface because of the long-range nature of coulombic interactions[14,15]. To gain insight into how $Ca^{2+}$ binding activates Xkr4, we performed a coarse-grained molecular dynamics (MD) simulation based on the modeled Xkr4 structure (Supplementary Fig. 7a, b). The $Ca^{2+}$-bridging effect was modeled by effectively enforcing a pseudo chemical bond between pairs of residues (D123, D127, and E310). We indeed observed changes in amino acids movement at local regions synchronously with the contact formation between TM1 and TM3 of aXkr4, suggesting that $Ca^{2+}$ binding on TM1 and TM3 induces the conformational change of Xkr4. Further work should expand this direction by performing atomistic MD simulations on the solved structures of Xkr4ΔC and Xkr4ΔC binding to XRCC4/C.

Compared to ubiquitously-expressed Xkr8, Xkr4 is strongly expressed in the brain[13] where PS exposure is involved in the removal of unwanted synapses in living neurons[32–35]. It is well known that extracellular $Ca^{2+}$ is incorporated in activated synapses to regulate neuronal function[36,37]. Due to the restricted size and limited access to the synaptic cleft, it was predicted that the influx of calcium from the presynaptic and postsynaptic regions during neurotransmission would greatly decrease the calcium concentration within the cleft, potentially reaching as low as 0.3 mM[38–42]. The actual measurement is necessary, but if extracellular calcium is decreased to less than 0.3 mM

at the active synapse, it might prevent the engulfment of active synapses by more efficiently suppressing Xkr4 activation.

In summary, our results demonstrate a critical role of $Ca^{2+}$ for the activation of Xkr4, Xkr8, and Xkr9, in which $Ca^{2+}$ functions as a molecular glue to bind to the negatively charged residues in TM1 and TM3 to connect the two transmembrane helices. Moreover, our present findings also suggest that XRCC4/C might bind to Xkr4 to induce an intermediate state for the association of $Ca^{2+}$ binding. We believe that these findings will contribute to understanding the mechanism of activation of the Xkr family and other scramblases.

## Methods

### Ethics statement
Animal study protocols were approved by Committee on the Animal Experiments of Kyoto University (Approved Number:49-4).

### Cell culture
HEK293T cells were maintained in Dulbecco's modified Eagle's medium (WAKO) containing 10% FBS (Gibco) and antibiotics (Nacalai). Mouse interleukin 3 (IL-3)-dependent Ba/F3 cells[43] were maintained in RPMI1640 (WAKO) containing 10% FBS, antibiotics, 55 μM β-mercaptoethanol, and 45 units/ml IL-3[44]. Human PLB985 cells (hereafter PLB)[45] were maintained in RPMI1640 containing 10% FBS, antibiotics, and 55 μM β-mercaptoethanol. HEK293T cells, Ba/F3 cells and PLB985 cells were grown at 37 °C in 5% $CO_2$.

## Construct preparation

Mouse (m) and human (h) Xkr4 full-length (Xkr4FL), or C-terminal cleaved form (Xkr4ΔC), were constructed as previously described[17]. Briefly, mXkr4 fused with a N-terminally V5 and C-terminally FLAG was cloned into the pMXs-puro vector[46]. plenti-IRES-puro vector harboring hXkr4, hXkr8, and hXkr9 tagged with a N-terminally V5 were constructed previously[17,47]. plenti-IRES-puro vector harboring V5-tagged mXkr4, mXkr8 and mXkr9 were also constructed. hXkr4, mXkr8 and hXkr8, and mXkr9 and hXkr9 tagged with a N-terminally V5 and C-terminally FLAG were introduced into the plenti IRES-puro vector[17]. mXkr4ΔC fusing with N-terminally SPOT-tag and C-terminally FLAG tag was cloned into lentiviral vector as described previously[17]. mXkr4ΔC Q332E (aXkr4), neutralized mutants of aXkr4 (D123N, D127N, E310Q), Ala mutants of aXkr4 (D123A, D127A, E310A, T307A, S311A, S339A), Cys single mutants of aXkr4 (D123C, V124C, G125C, T126C, D127C, E310C), Cys double mutants of aXkr4 (D123C/E310C, V124C/E310C, G125C/E310C, T126C/E310C, D127C/E310C), Lys and Arg mutants of aXkr4 (D123K, D123R, G125K, G125R, D127K, D127R, E310K, E310R), Xkr4FL E310K, and Xkr4ΔC E310K were cloned into the pMXs vector with a V5 tag to their N-terminus and a FLAG tag and tagRFP to their C-terminus. mXkr4ΔC was also inserted into pNEF vector[12] with a V5 tag to its N-terminus and an HRV3C-cleavage site, and FLAG and monomeric EGFP to its C-terminus. The mutant MFG-E8 D89E[48] with a C-terminus monomeric EGFP and 8×His-tag was cloned into plenti vector and a 3 × GGGS linker was fused at N-terminus and C-terminus of monomeric EGFP. The XRCC4 fused with a C-terminal tagRFP was cloned into plenti vector as described previously[17].

In order to express XRCC4 in bacteria, the cDNA of full-length human XRCC4 was codon optimized and synthesized by Sangon Biotech and cloned into the pGEX-5X-1 vector. Point mutations (I266G or R270A) were inserted into the C-terminal fragment of XRCC4 (XRCC4/C, 265–366). The enterokinase (EK) recognition site Asp-Asp-Asp-Asp-Lys was fused at the N-terminus and an 8×His-tag was inserted at the C-terminus of the XRCC4/C. The cDNA of bovine EK catalytic subunit (cEK) was codon optimized and synthesized by Sangon Biotech, and the EK recognition site was added in the N-terminus of cEK, and cloned into the pGEX-5X-1 vector.

## Establishment of cell lines

Ba/F3 cells deficient in *TMEM16F* and *Xkr8* (BDKO) were previously established[17]. Retroviruses and Lentiviruses were produced as described[17]. Briefly, the retroviral vector pMXs or pMXs-puro, pGag-pol IRES bsr (gifted from Dr. Toshio Kitamura) and pCMV-VSVG (RIKEN) were transfected into HEK293T cells in a 10 ml medium by polyethylenimine MAX (PEI) (Polysciences). Five ml fresh medium was added into cells in 24 h after transfection. The retroviruses in supernatant after culturing for 48 h were collected, filtered, and harvested by centrifugation at 6000 × g for 16 h. Then, retroviruses were resuspended in 500 μl fresh medium containing 10 μg/ml polybrene and used to infect 5 × 10^5 BDKO cells. Lentiviruses were produced in HEK293T cells by transfecting the lentiviral vectors plenti, pCMV-VSVG-RSV-Rev (RIKEN) and pCAG-HIVgp (RIKEN) with PEI. Cells were supplemented with 5 ml fresh medium in 24 h after transfection. The supernatant containing lentiviruses was collected, filtered, and enriched by centrifugation at 6000 × g for 16 h after transfection for 48 h, and used to infect 5 × 10^5 PLB cells (gifted from Prof. W. Hiraoka) with 10 μg/ml polybrene. CHO cells (gifted from Prof. Kengaku) expressing MFG-E8 D89E with a C-terminus monomeric EGFP and 8 × Histidine were also established by lentivirus infection.

## Expression of active cEK

The cEK protein was expressed and purified from *Escherichia coli* (*E. coli*) BL21(DE3). Briefly, BL21(DE3), transformed with the cEK plasmid, was cultivated in LB medium at 37 °C up to the OD at 0.6–0.8. Then, 0.5 mM IPTG was added to induce cEK expression and cells were

cultured at 37 °C for 4 h. Bacteria were collected by centrifugation (6000 × g, 4 °C, 5 min) and lysed by sonication in lysis buffer (25 mM Tris-HCl (pH 7.5), 0.5 mM EDTA). The insoluble fractions containing inclusion bodies were harvested by centrifugation (15,000 × g, 4 °C, 20 min). The insoluble fractions were washed with 0.9% Triton X-100 one time and washed with 0.5% NaCl twice. The washed inclusion bodies were dissolved in denaturation buffer (50 mM Tris-HCl (pH 9.0), 10 mM EDTA, 8 M urea) for 1 h at room temperature, and further incubated for 1 h at room temperature with 10 mM β-mercaptoethanol. The denatured sample was diluted in renaturation buffer (50 mM Tris-HCl (pH 9.0), 1 mM glutathione disulfide (GSSG), 3 mM glutathione (GSH)) and incubated at 25 °C for 2 days. The activated cEK was collected and stored at 4 °C for a few days until use.

## Expression and purification of XRCC4 and mutants

The proteins of full-length XRCC4 (WT), XRCC4/C, and its point mutants XRCC4/C I266G and XRCC4/C R270A were expressed in BL21(DE3). Bacteria were grown at 37 °C in LB medium while shook at 150 rpm up to the OD at 0.6–0.8. and incubated with 0.5 mM IPTG for overnight at 16 °C for XRCC4/C proteins and 15 °C for WT to induce gene expression. Bacteria were harvested at 6000 × g, 4 °C, 5 min, and lysed by sonication in lysis buffer (PBS containing 10 mM imidazole). The insoluble materials were removed from the lysate by centrifugation (20,000 × g, 4 °C, 20 min). The clear supernatant was filtered and applied to glutathione Sepharose (GE healthcare), equilibrated with lysis buffer. After washing with lysis buffer, the proteins were eluted with elution buffer (50 mM Tris-HCl (pH 8.0), 10 mM GSH). To cleave the GST tag in the N-terminus of XRCC4/C and mutants, the eluted samples were incubated at 4 °C overnight with cEK at a molar ratio of 1:8 (XRCC4/C:cEK) in cleavage buffer (20 mM Tris-HCl (pH 8.0), 200 mM NaCl, 2 mM CaCl₂). To isolate the cleaved protein, the protein mixture was passed over nickel column (FUJIFILM), equilibrated with wash buffer (10 mM HEPES (pH 8.0), 0.5 M NaCl, 10 mM imidazole). After washing with wash buffer, proteins were eluted with elution buffer (10 mM HEPES (pH 8.0), 150 mM NaCl, 200 mM imidazole). The eluate was further purified using a HiLoad 16/600 Superose 75 pg (AKTA purifier FPLC system, cytiva), where the peak corresponding to XRCC4/C was collected into the tubes and further concentrated using an Amicon Ultra Centrifugal filter (3 kDa MWCO, Merck). WT was purified by nickel column and further purified by size exclusion chromatography. All purification steps were carried out under 4 °C, and the purified proteins were flash frozen using liquid nitrogen and stored at −80 °C until use.

## Expression and purification of MFG-E8

To detect PS exposure without extracellular Ca²⁺, a recombinant MFG-E8 D89E fused with 3 × GGGS linker, monomeric EGFP, 3 × GGGS linker, and 8×Histidine, was expressed in CHO cells by the lentiviral system. After infection, highly MFG-E8-expressing cells, corresponding to top 0.5% of GFP-positive cells, were sorted by flow cytometry (FACS AriaII). The sorted cells were seeded into 300 ml of DMEM/Ham's F-12 (Wako) medium containing 10% FBS (Gibco) and penicillin-streptomycin mixed solution (Nacalai). One day later, the cells reached to 80% confluent, and the medium was changed to a fresh medium containing antibiotics, but not FBS. The cells were cultured for additional four days, with a medium collection performed every two days. The collected medium was mixed with 0.1% TritonX-100 and 5% glycerol, and applied for purification with Ni-NTA agarose resin (Wako). The protein was eluted with 300 mM imidazole, 25 mM Tris (pH 8.0), 150 mM NaCl, and concentrated using Amicon® Ultra (50 kDa cut off) into HBSS buffer. After concentration, 0.1% BSA was added as a carrier and used for the assay of PS exposure.

## Whole lysate preparation

pNEF Xkr4ΔC vector was expressed in HEK293T cells by transient transfection with PEI at a 3:1 ratio of PEI and plasmid. Cells were

washed twice with cold PBS, collected with scraper, and incubated with PBS containing 1 mM indicated metal ions or 0.5 mM EGTA on ice for 20 min. Cells were collected by centrifugation at $400 \times g$, 4 °C, 2 min, flash frozen with liquid nitrogen, and stored at −80 °C until use. To prepare the whole cell lysate, cell pellets were solubilized in solubilization buffer (10 mM HEPES-NaOH (pH 7.5), 100 mM 6-aminocaproic acid, 140 mM NaCl, 1% detergent, 10% glycerol, 1 mM indicated metal ions or 0.5 mM EGTA, 1 mM β-APMSF, protease inhibitor cocktail set V (EDTA free) (FUJIFILM), 1 mM NaF on ice for 1 h. Supernatant was collected by centrifugation (20,000 × $g$, 4 °C, 20 min).

### Collection of Xkr4 dimer fraction
pNEF Xkr4ΔC or aXkr4 fused with monomeric EGFP vectors were transfected in HEK293T cells with PEI, and the whole lysate was prepared as described above. For collection of Xkr4ΔC and aXkr4 dimer fraction, the whole lysate was loaded on a size exclusion chromatography (SEC) column (Superdex Increase 200 10/300 GL, GE Healthcare), which was equilibrated with a column buffer (10 mM HEPES-NaOH (pH 7.5), 150 mM NaCl, 0.01% LMNG/GDN at a ratio of 1:1, 10% glycerol, 1 mM CaCl$_2$ or 0.5 mM EGTA). SEC analysis was performed at 0.35 ml/min, and fractions containing Xkr4ΔC or aXkr4 dimer were collected at 39 to 40 min according to the fluorescence intensity. Pooled samples were flash-frozen using liquid nitrogen and stored at −80 °C until use.

### Thermostability assay
The fluorescence intensity of GFP derived from collected Xkr4ΔC or aXkr4 dimer was measured using a fluorescent microplate reader and adjusted into the same relative fluorescence unit (RFU) of 2000. Diluted samples were incubated at 4, 16, 25, or 37 °C for 1 h. The supernatant was collected by centrifugation (20,000 × $g$, 4 °C, 20 min), adjusted into fluorescence intensity at 800 RFU, and samples were applied to Blue Native (BN)-PAGE. The protein level was quantified by ImageJ (https://imagej-nih-gov.kyoto-u.idm.oclc.org/ij/), and the total amount of dimer and monomer in an experiment at 4 °C was used to define as 100%.

### BN-PAGE
BN-PAGE was performed as previously described[17]. Briefly, collected samples were diluted with column buffer and applied to NativePAGE Novex 4%-16% (wt/vol) Bis-Tris gels (Thermo Fisher Scientific). Electrophoresis was performed at 150 V for 30 min at 4 °C in 0.02% CBB G-250, followed by further electrophoresis at 150 V for 120 min in 0.002% CBB G-250. The gel was incubated with SDS running buffer (20 mM Tris-HCl (pH 8.3), 190 mM Glycine, 0.1% SDS) at RT for 20 min, transferred to the PVDF membrane (Sigma) at 0.1 A for 1 h, and then applied to western blotting. To detect the desired protein in the PVDF membrane, the PVDF membrane was blocked with 5% skim milk in TBS-T (25 mM Tris-HCl (pH 7.5), 150 mM NaCl, 0.05% Tween 20 (Bio-Rad)) for 3 h and incubated with anti-V5-HRP antibody (1/6000 dilution, Thermo Fisher Scientific).

### Apoptotic stimulation with staurosporine (STS)
PLB cells expressing desired proteins were seeded with fresh medium one day before analysis. Next day, PLB cells were resuspended in pre-warmed fresh medium at $1 \times 10^6$ cells/ml and cultured at 37 °C for 1 h. STS was added to the cells at a final concentration of 10 μM. Cells were further incubated at 37 °C incubator for 3 h.

### Thapsigargin treatment
BDKO cells expressing aXkr4 were washed and resuspended in a pre-warmed medium. Cells were incubated at 37 °C in 5% CO$_2$ for 1 h, followed by incubation with a final concentration of 4 μM Fluo4-AM (Dojindo) for 30 min and then with a final concentration of 1 μM

thapsigargin (Wako) for 20 min. For analysis of the intracellular Ca$^{2+}$, cells were collected, washed with pre-chilled HBSS buffer, resuspended in 200 μL HBSS containing with or without 0.5 mM EGTA, and measured by flow cytometry. For PLS activity assay, thapsigargin-treated cells were incubated with NBD-PC and analyzed with flow cytometry.

### Apoptotic stimulation with ultraviolet (UV) irradiation
Apoptotic stimulation was performed as described[17]. Briefly, PLB cells were resuspended in PBS, and then UV-irradiated at 2000 J/m$^2$ (CL-1000(253 nm) crosslinker) for cells expressing Xkr4, or at 200 J/m$^2$ for cells expressing Xkr8 or Xkr9. The irradiated cells were collected and cultured with pre-warmed fresh medium at 37 °C for 3 h (Xkr4) or 2 h (Xkr8 and Xkr9), and then applied to the PLS assay.

### Electroporation of purified XRCC4
The purified XRCC4 WT and XRCC4/C were stored in PBS. To perform electroporation, one million BDKO cells expressing mXkr4 or mXkr4ΔC were washed with opti-MEM (Gibco) and resuspended in 100 μl opti-MEM. Final concentration at 0.5 μM of purified XRCC4 WT or XRCC4/C was added into cell suspension and mixed well. Electroporation was performed as described[17]. The electroporated cells were cultured with 2 ml RPMI 1640 containing 10% FBS at 37 °C for 1 h. Half a million cells were collected and applied to the PLS assay.

### Cross-link assay
BDKO cells expressing each Cys mutant were washed with TBS buffer (25 mM Tris-HCl (pH 7.5), 140 mM NaCl). Cells were then incubated with a final concentration of 1.1 mM copper-phenanthroline (CuP) (Tokyo Chemical Industry), dissolved in 20% ethanol, in TBS buffer at room temperature for 20 min or incubated with a final concentration of 25 μM iodine solution (Wako) in TBS buffer at room temperature for 10 min. Cells were collected by centrifugation at $400 \times g$, 4 °C, 2 min, and applied to the PLS assay.

### PLS assay
The PLS assay was performed using NBD-PC, NBD-sphingomyelin (NBD-SM) (Avanti Polar Lipids) or purified MFG-E8-GFP. Half a million cells were collected and washed with chilled HBSS buffer, and then resuspended in 200 μl HBSS buffer containing a defined concentration of Ca$^{2+}$ or EGTA. After 10 min incubation on ice, the cells were mixed with 200 μl HBSS buffer containing a final concentration of 0.2 μM NBD-PC or 0.2 μM NBD-SM and incubated on ice for 12 min for NBD-PC, or 40 min for NBD-SM. A 5 mg/ml fatty acid-free BSA in HBSS buffer containing 1 μg/ml DAPI was added to remove the cell surface NBD-PC and incubated on ice for more than 2 min. PLS activity was analyzed by flow cytometry. For A23187 (Sigma) treatment, a defined concentration of A23187 was added to the cell suspension in NBD-PC-containing buffer, incubated on ice for 4 min, and analyzed by flow cytometry.

MFG-E8-GFP was used to detect PS exposure. Half a million cells were collected and washed with chilled HBSS buffer, and then resuspended in 50 μl HBSS buffer containing a defined concentration of Ca$^{2+}$ and incubated on ice for 10 min. Purified MFG-E8-GFP was then added and incubated on ice for 40 min. The final concentration of 1 μg/ml DAPI was added in HBSS buffer to remove necrotic cells during analysis. PS exposure was analyzed by flow cytometry. All of flow cytometry analysis were performed using a FACSLyric (BD Biosciences) or FACS ARIA2 (BD Biosciences).

### Engulfment by phagocytes
Engulfment was performed as previously described[17]. Briefly, C57BL/6 J, 8-week-old female mouse was injected with 2 ml of 3% thioglycolate into the peritoneum. After 3 days, cells were collected from the mice

peritoneum with DMEM containing 10% FBS. To induce apoptosis, PLB cells expressing SPOT-Xkr4-FLAG and XRCC4-tagRFP were treated with UV, and labeled with 0.1 µg/ml pHrodo Green STP ester (Invitrogen). Apoptotic cells were incubated with thioglycolate-elicited peritoneal macrophages at 37 °C for 4 h. Then, macrophages were detached by trypsin/EGTA (Nacalai) at 37 °C for 3 min. Detached macrophages were incubated with 400 fold-diluted APC-labeled anti-CD11b antibody (BioLgend) at 4 °C for 20 min in CHES buffer (20 mM CHES pH 9.0, 150 mM NaCl, and 2% dialyzed FBS). Finally, macrophages were washed and resuspended in analysis buffer (20 mM CHES pH 9.0, 150 mM NaCl) containing 0.25 µg/ml DAPI and analyzed by flow cytometry.

### Membrane tension measurements

Membrane tension measurement was performed as previously described[49]. Here, BDKO cells expressing aXkr4 or aXkr4 E310A mutant were incubated with 0.1 µg/ml anti-mouse basigin (Biolegend) in HEPES buffer (10 mM HEPES (pH 7.5), 140 mM NaCl, 1% BSA) at 4 °C for 45 min, then washed with HEPES buffer for 2 times, and incubated with 0.2 µg/ml anti-rat IgG biotin (Thermo Fisher Scientific) at 4 °C for 45 min. Cells were seeded on a glass bottom dish (IWAKI) which was pre-coated with 0.05 mg/ml poly-L-lysine, and incubated with streptavidin conjugated beads (SPHERO™). To examine the membrane tension, cells were cultivated in DMEM, which does not contain $Ca^{2+}$, in the presence of 0 mM, 2 mM, or 5 mM $Ca^{2+}$ at RT to analyze membrane tension.

Cell-attached beads were trapped and pulled using an optical trap system (MMS-1064-200-2L/2E/2 S, Sigma Koki) with a 1064-nm laser. A 100× oil objective (UPLSAPO100XO, NA = 1.4, Olympus) mounted on an inverted microscope (IX71, Olympus) was used to trap beads and visualize cells and beads in bright field. Beads attached on the cell membrane were pulled away from the stationary laser trap via a motorized stage (BIOS-225T, Sigma Koki) and images were captured for the membrane tension analysis with a CCD camera (Zyla 5.5 cMOS, Andor) every 10 ms. Using these images, force-extension curves were acquired by calculating the force $F$ exerted on the trapped beads as $F = K_x \Delta x$, where $\Delta x$ is the deviation of the trapped bead from its original position and the spring constant of the optical trap $K_x = 36.55[pN/\mu m]$. Membrane tension was obtained by applying a linear fit to the force-extension curves, which was performed using matplotlib 3.5.1 and numpy.

### Prediction of mXkr4 structure

Main structure (including TMs) of mXkr4 structure was predicted using RoseTTAFold[25] (https://robetta.bakerlab.org/submit.php), and the loop structure of mXkr4 was optimized by Modeler 10.1. All the views of structures were analyzed using the chimera 1.16 or chimeraX 1.4 software.

### Molecular dynamic (MD) analysis

All simulations were performed using CafeMol 3.2.1[50]. The AICG2+ model was used for protein analysis. In the coarse-grained model, a bead was used to represent each amino acid at the Cα atomic position. Electrostatic and excluded volume interactions were used to model interactions between beads. Langevin dynamics was used for time propagation. The temperature was set to 300 K. The dielectric constant was set to 78.0. The monovalent ion concentration was set to 150 mM. The structure of aXkr4 was predicted by RoseTTAFold Server[25]. To stimulate the "calcium bridge", we applied a 'bridging effect' between each pair of postulated primary calcium-binding residues (D123, D127, and E310), based on Hooke's law $E(r) = -k (r - r_0)^2$ (if $r < r_0$), where $k$ is the force constant, $r$ is the distance between the residues, and $r_0$ the equilibrium distance. This 'bridging effect' was enforced by setting $r_0$ to 5.0 Å and $k$ to 5.0 J/Å$^2$.

### Data analysis

Data analysis for flow cytometry analysis was performed using FlowJo (FlowJo LLC), and the dose response-stimulation analysis was performed using prism (Graphpad). The calculation formula is as described below:

$$F/F \max = \frac{100X}{EC50 + X}$$

Where $F$ means the fluorescence value of incorporation of NBD-PC at each concentration of $Ca^{2+}$. $F$ max is the maximum fluorescence among a series of $Ca^{2+}$ concentration. $X$ is the corresponding $Ca^{2+}$ concentration. The unit for $EC50$ is the same as one for $X$.

### Reporting summary

Further information on research design is available in the Nature Portfolio Reporting Summary linked to this article.

## Data availability

The Genbank accession codes for mouse Xkr4 and human Xkr4, mouse Xkr8, human Xkr8, mXkr9 and human Xkr9 are NM_001011874, NM_052898, NM_201368, NM_018053, NM_001011873, and NM_001011720, respectively. Previously published cryo-EM models of mXkr8 and rXkr9 used in this study are available in the Protein Data Bank (PDB) under the PDB ID 7DCE and 7P16, respectively. All the data produced or analyzed in this study has been incorporated into this article and its supplementary files. Source data are provided with this paper. The MD simulation codes have been provided with this paper as Supplementary Software. Source data are provided with this paper.

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

## Acknowledgements

The authors thank A. Fujimoto for secretarial assistance, E. Alvi for critical editing, and the Suzuki lab members for discussion. This work was supported by AMED-PRIME (15665392), AMED-FORCE (19191470), Grants-in-Aid for Scientific Research on Innovative Areas "Dynamic regulation of brain function by Scrap & Build system" (KAKENHI 16H06456), JST-CREST (1199566), Joint Usage and Joint Research Programs of the Institute of Advanced Medical Sciences of Tokushima University, WPI-iCeMS, Takeda Science Foundation to J.S., and MEXT Grants-in-Aid for Scientific Research (C) (KAKENHI 20K06486) to M.M.

## Author contributions

P.Z. and J.S. designed the overall research and interpreted experimental results. P.Z. performed most of the experiments. M.M. found the requirement of $Ca^{2+}$ for Xkr4 activation. R.S. and M.T. performed and

interpreted membrane tension experiments. Hikaru.K. analyzed some of Xkr4 mutants and performed Molecular dynamic analysis with D.P. Y.D. performed a purification of MFG-E8 fused with monomeric EGFP. Hidetaka.K. performed protein analysis with P.Z. P.Z., and J.S. wrote the manuscript.

## Competing interests

The authors declare no competing interests.
