## [Peer Review File · Nature Communications]

REVIEWER COMMENTS

Reviewer #1 (Remarks to the Author):

In their manuscript, the authors have used a variety of cellular assays to investigate the transport properties of the lipid scramblase XKR4, which is activated by the proteolytic cleavage of the C-terminus by Caspase-3 in response to the induction of apoptosis. The described work follows up on a previous study by the authors, where they have investigated the activation mechanism of the protein, which involves next to a dimerization induced by the cleavage of the C-terminus, also the interaction with a peptide that is released during the processing of the protein XRCC4 by caspases.

In the current work, the authors describe the role of Ca²⁺ in the activation process. In a set of elegant and conclusive experiments, they show that Ca²⁺, acting from the extracellular side, is an essential cofactor of the activation process in XKR4. A similar role was also found for the paralogs XKR8 and 9, which are both also activated by Caspase-3 by an equivalent mechanism. Based on sequence alignments and the analysis of known structures of XKR8 and 9 and a model of XKR4, they have identified candidates of a putative Ca²⁺-binding site, located on α -helices 1 and 3. Both helices bury conserved acidic residues in their interface. By mutagenesis, the authors identify three close-by residues, whose mutation exerts a strong influence on the activation process by disturbing the Ca²⁺ interaction and in that way prevent activation. In a set of complementary experiments, they were able to activate a construct containing two close-by cysteine residues by the formation of a disulfide bond in the absence of Ca²⁺ thus further emphasizing the importance of Ca²⁺ in stabilizing an active protein conformation.

This is an excellent manuscript showing data of high quality that provide important insight into the activation mechanism of a central, yet still poorly understood molecular process. The results are intriguing for several reasons. Although structures of XKR family members have become available recently, the mechanism of how this important family catalyzes passive and bidirectional lipid transport between both leaflets of the membrane has remained elusive. Additionally, it was so far impossible to demonstrate lipid scrambling for any of the caspase-activated scramblases of the XKR family in a purified and reconstituted system. In case of the unrelated TMEM16 family, this was a key experiment to settle the controversy whether these proteins carry the catalytic unit to facilitate lipid flip-flop.

Retrospectively, and in light of the described results, the absence of pronounced conformational changes in the structure of the caspase-cleaved XKR9 and the lack of activity in a reconstituted system can be attributed to the absence of Ca²⁺ in the buffers. While, due to the continuously high extracellular Ca²⁺-concentrations, a regulatory role of the divalent cation appears unlikely, the role as stabilizing cofactor is entirely credible given the high quality and robustness of the data and the location of the identified residues in known structures.

I am convinced that the manuscript will be of large importance for the field and thus consider it as strong candidate for publication in Nature Communications. I do not think that the study requires more data and only have few remarks that might improve the clarity of the work:

Activation Mechanism of XKR4:

- The authors have investigated whether the formation of a crosslink between $\alpha 1$ and $\alpha 3$ would be sufficient to activate the uncleaved protein or the ΔC construct without the peptide XRCC4/C and found no activity in the former and less activity in the latter case. Based on these results they conclude that the formation of the disulfide bridge mimicking the binding of Ca^{2+} would be insufficient to activate the protein. I wonder whether they have investigated the efficiency of disulfide bond formation in these cases as it might well be that the presence of the C-terminus and absence of the activating peptide would stabilize a conformation that would strongly disfavor disulfide bond formation.

- Along the same lines, is there any evidence that Ca^{2+} would be bound to the protein already in the inactive or partially activated state?

- In general, the experimental data is intriguing and of very high quality. The weakest piece of evidence in my opinion concerns the blue native gel data shown in Extended Data Fig. 4, which is used to claim that Ca^{2+} binding does not affect dimer stability. While I agree that the data provides qualitative evidence for such behavior, the appearance of a band with faster migrating behavior at 25 °C is apparent in the sample containing EGTA and could indicate the dissociation of the dimer in absence of divalent cations already at room temperature.

Reviewer #2 (Remarks to the Author):

Phospholipid scrambling is an important cell signaling process that remains incompletely understood. Two families of phospholipid scramblases have been identified; TMEM16s and XKRs. Generally, it has been thought that the TMEM16 scramblases are activated by increases in cytosolic Ca and the XKR scramblases are activated by caspase cleavage. This manuscript makes an important contribution by showing that in addition to the requirement for caspase activation, XKR proteins also require extracellular Ca for their scramblase activity. The authors also identify a Ca binding site in the transmembrane region of XKR4, XKR8, and XKR9 that is accessible to the extracellular space. This discovery adds a new dimension to XKR scramblase that may have important implications for understanding the regulation and function of phospholipid scramblases. This is a very nice study with strong data, but there are a few points that need additional development and strengthening.

Major Comments:

(1) Role of intracellular Ca:

The authors suggest that intracellular Ca is not required because ionomycin does not enhance XKR activity. This is not convincing because the question is whether intracellular Ca in the absence of extracellular Ca stimulates XKR4. The best experiment would be to have EGTA extracellular and elevate intracellular Ca with thapsigargin or caged-calcium while measuring cytosolic Ca. This may be a difficult experiment in practice. Another approach would be to buffer intracellular Ca with BAPTA-AM, but the authors should measure intracellular Ca during their manipulations.

(2) Divalent cation selectivity:

The authors should also test Ba²⁺ and Tb³⁺. Ba²⁺ is a very important divalent ion to test because it can distinguish between different kinds of Ca²⁺ binding sites. For example, Ba²⁺ binds with different affinity to the two lobes calmodulin, and while Ba²⁺ is more conductive than Ca²⁺ through L-type Ca channels, Ba²⁺ does not support Ca²⁺ - dependent channel inactivation. Tb³⁺ binds very tightly to many Ca²⁺ binding sites in proteins, but in some proteins it acts as an activator and others as an inhibitor. Further, Tb³⁺ fluorescence can be used to study protein conformational changes, so these data could be useful to other investigators for structural studies. I am surprised that Mn²⁺ apparently binds (although at very low affinity). But this raises a possible technical question. Unless great care is taken, even >18MΩ deionized water can contain ~100 μM Ca²⁺. And most reagent grade divalent cations contain additional trace Ca contamination. How can the authors rule out Ca contamination of the Mn solution?

(3) Ca binding site:

The evidence the authors provide for the Ca²⁺ binding site is strong. But yet, there are certain features of the binding site that seem odd. First, I am surprised that Mn binds even at low affinity. It suggests the binding site is quite flexible since Mn has a much smaller ionic radius than Ca.

Second, the binding residues are too far apart (6 -9Å) in the model to coordinate Ca. This might be expected, because in the absence of Ca, they are repelling one another. But also, this might be a shortcoming of the homology model because the corresponding residues in 7DCE are closer than they are in the XKR4 homology model. Also, the Alpha-Fold model of mXKR4 shows these residues being closer together than the authors' homology model. It might be worthwhile running an MD energy minimization on the XKR4 model (also with Ca added) to see if this improves the binding site configuration. The authors do not provide much detail in their methods how the homology model was constructed or whether there are any probability estimates of confidence of their model. Knowing the architecture of the binding site is important because a 4 – 5 Å rearrangement of the binding site could produce substantial conformational changes elsewhere in the protein that are important to consider.

Typically Ca^{2+} is stabilized by 6 - 8 oxygen atoms. The side chains of D123, D127, and E310 may account for 4 contacts, but where are the others? The others could be provided by waters or backbone carbonyls, but it would be nice to have a model of how this binding site would look and how it might change the conformation of the protein.

(4) Cross-linking:

While the cross-linking experiments are suggestive, it is disappointing that the D123C – E310C mutants did not work as expected. Another experiment that might help convince the readers would be to make the D127K or D123K mutants (or D123R or D127R). One would expect these to form salt bridges with E310 if they are close enough to coordinate Ca.

(5) Physiological significance

Because extracellular Ca is physiologically rather constant at ~2 mM, changes in Ca occupancy are unlikely to play a physiological role in XKR4 activity unless the affinity of this site is modulated by another mechanism. That is not to say that Ca binding to this site is not important to XKR4 function, but the authors should be more precise how they envision the physiological significance of this binding site.

(6) Language: The writing could be improved to improve clarity and grammar. A few examples are given below, but these are not comprehensive and the authors should edit their language throughout the manuscript.

The sentence on page 2 “Using three types....” is too convoluted with too many phrases that are hard to parse. The next sentence is incomplete.

Page 3 “We induced cell death with” Would be more clear if it read “in cells expressing either the FL or delta-C form.”

Page 3 “was applied to our PLS activity”. This phrase is nonsense.

Page 4 “where negatively charged amino acids were pronounced” is awkwardly phrased.

Page 4 “to remove the effects of the side chain”. Ala does have a side chain, so this phrasing is odd.

Page 6. “Single Cys mutants in TM1 with or without E310C” seems awkward because E310C is already a single mutant. This could be rephrased.

Page 6. “lesser” is incorrect

Reviewer #3 (Remarks to the Author):

Zhang et al. have shown that Xkr4 requires extracellular Ca^{2+} as an additional factor to become a functional lipid scramblase by performing a series of experiments including optical tweezers, structural prediction and biochemical experiments. They also uncovered potential Ca^{2+} binding sites (e.g., D123 and D127 in TM1 and E310 in TM3) in Xkr4. Consistently, its scramblase activity was absent in mutants that neutralize these acidic residues. The present manuscript is very interesting and contains some novel findings, as described above, with supporting biochemical and biophysical data based on their strong expertise. However, additional important aspects need to be addressed for publication in Nature Communications. The authors need to explain in detail the physiological meaning of extracellular Ca^{2+} -mediated Xkr4 functions. For example, it is essential to show whether extracellular Ca^{2+} acts as a molecular switch for the physiological and/or pathophysiological functions of Xkr4. It is also important to prove the structural changes via coordination of Ca^{2+} in the TM1 and TM3 of Xkr4 by performing intensive structural analysis. Therefore, this manuscript is not suitable for publication in this esteemed journal in its present form.

Additional comments:

Although the authors conclude that Xkr4 activity regulates membrane tension, the molecular mechanism is not fully addressed in the present manuscript. To strengthen the significance of this manuscript, it is important to show whether Xkr4-mediated phospholipid scrambling directly regulates intrinsic membrane properties or whether it indirectly affects membrane tension via the rearrangement of the intracellular cytoskeleton.

In optical tweezer experiments, why did the authors use an external solution containing 5 mM Ca^{2+} ? To evaluate physiological membrane properties, the concentration of Ca^{2+} should be approximately 2 mM.

To evaluate the role of external Ca^{2+} in Xkr4 function as a phospholipid scramblase in detail, the authors are encouraged to measure Xkr4-mediated translocation of other phospholipids, such as phosphatidylserine and phosphatidylethanolamine.

Our point-by-point responses to the reviewers' comments are described below.

Reviewer #1: *In their manuscript, the authors have used a variety of cellular assays to investigate the transport properties of the lipid scramblase XKR4, which is activated by the proteolytic cleavage of the C-terminus by Caspase-3 in response to the induction of apoptosis. The described work follows up on a previous study by the authors, where they have investigated the activation mechanism of the protein, which involves next to a dimerization induced by the cleavage of the C-terminus, also the interaction with a peptide that is released during the processing of the protein XRCC4 by caspases.*

In the current work, the authors describe the role of Ca²⁺ in the activation process. In a set of elegant and conclusive experiments, they show that Ca²⁺, acting from the extracellular side, is an essential cofactor of the activation process in XKR4. A similar role was also found for the paralogs XKR8 and 9, which are both also activated by Caspase-3 by an equivalent mechanism. Based on sequence alignments and the analysis of known structures of XKR8 and 9 and a model of XKR4, they have identified candidates of a putative Ca²⁺-binding site, located on α -helices 1 and 3. Both helices bury conserved acidic residues in their interface. By mutagenesis, the authors identify three close-by residues, whose mutation exerts a strong influence on the activation process by disturbing the Ca²⁺ interaction and in that way prevent activation. In a set of complementary experiments, they were able to activate a construct containing two close-by cysteine residues by the formation of a disulfide bond in the absence of Ca²⁺ thus further emphasizing the importance of Ca²⁺ in stabilizing an active protein conformation.

This is an excellent manuscript showing data of high quality that provide important insight into the activation mechanism of a central, yet still poorly understood molecular process. The results are intriguing for several reasons. Although structures of XKR family members have become available recently, the mechanism of how this important family catalyzes passive and bidirectional lipid transport between both leaflets of the membrane has remained elusive. Additionally, it was so far impossible to demonstrate lipid scrambling for any of the caspase-activated scramblases of the XKR family in a purified and reconstituted system. In case of the unrelated TMEM16 family, this was a key experiment to settle the controversy whether these proteins carry the catalytic unit to facilitate lipid flip-flop. Retrospectively, and in light of the described results, the absence of pronounced conformational changes in the structure of the caspase-cleaved XKR9 and the lack of activity in a reconstituted system can be attributed to the absence of Ca²⁺ in the buffers. While, due to the continuously high extracellular Ca²⁺-concentrations, a regulatory role of the divalent cation appears unlikely, the role as stabilizing cofactor is entirely credible given the high quality and robustness of the data and the location of the identified residues in known structures.

I am convinced that the manuscript will be of large importance for the field and thus consider it as strong candidate for publication in Nature Communications. I do not think that the study requires more data and only have few remarks that might improve the clarity of the work:

We appreciate the reviewer's encouraging comment: "I am convinced that the manuscript will be of large importance for the field and thus consider it as strong candidate for publication in Nature Communications."

-Activation Mechanism of XKR4:

The authors have investigated whether the formation of a crosslink between $\alpha 1$ and $\alpha 3$ would be sufficient to activate the uncleaved protein or the ΔC construct without the peptide XRCC4/C and found no activity in the former and less activity in the latter case. Based on these results they conclude that the formation of the disulfide bridge mimicking the binding of Ca^{2+} would be insufficient to activate the protein. I wonder whether they have investigated the efficiency of disulfide bond formation in these cases as it might well be that the presence of the C-terminus and absence of the activating peptide would stabilize a conformation that would strongly disfavor disulfide bond formation.

Response 1: We understand the reviewer's concern. Indeed, we performed mass spectrometry analysis to detect the disulfide bond formed in aXkr4 G125C/E310C after CuP as well as I_2 treatment. However, we could not detect any fragments derived from transmembrane domains because their hydrophobicity makes ionization of the peptides difficult. Instead, as described in **Response 7 to reviewer #2's comment**, we tried to induce a salt bridge in the Ca^{2+} -binding site of aXkr4 by performing a scan of the positively-charged amino acids. As a result, we found that strong PLS activity was induced without Ca^{2+} when E310 was substituted to Lys in aXkr4, but not in Xkr4FL or Xkr4 ΔC (shown below and also as Supplementary Fig. 6b in the revised manuscript), which is consistent with the result of disulfide bond formation (Fig. 6b, c). Also as described in **Response 2 and 3 to reviewer #1's comment**, Ca^{2+} stabilized the partially active state of Xkr4 (aXkr4) more than the inactive Xkr4 dimer (Xkr4 ΔC) (shown below and also as Supplementary Fig. 5a-d in the revised manuscript). Combining these results, we prefer a model that Ca^{2+} binds to the pre-activated state of Xkr4.

Supplementary Figure 6: Xkr4 activation by induced disulfide bond or salt bridge. b Xkr4 activation in Lys mutation. BDKO cells expressing aXkr4, aXkr4 E310K, Xkr4FL E310K, or Xkr4 ΔC E310K were applied to PC uptake assay without Ca^{2+} .

Supplementary Figure 5: Xkr4 stability. **a-d** Thermostability of Xkr4. Cells expressing V5 and EGFP-fused Xkr4ΔC (**a, b**) or aXkr4 (**c, d**) were solubilized with 0.5% GDN and 0.5% LMNG in the presence of 1 mM Ca²⁺ or 0.5 mM EGTA. The dimer fraction of Xkr4 was collected by size exclusion chromatography, incubated at 4, 16, 25, and 37 °C for 1 hr, and then applied to BN-PAGE, followed by detection of Xkr4 with anti-V5 antibody (**a, c**). The percentage of monomers was calculated and shown as averages of triplicates on the right with SD (**b, d**). ns, not significant. **, $p < 0.01$.

- Along the same lines, is there any evidence that Ca²⁺ would be bound to the protein already in the inactive or partially activated state?

Response 2: This is a significant point to be clarified. We compared the thermostability of Xkr4ΔC and aXkr4 in the presence of 1 mM Ca²⁺ or 0.5 mM EGTA to check whether Ca²⁺ binding affects protein stability. Although there are no significant differences between Ca²⁺ and EGTA for Xkr4ΔC (Supplementary Fig. 5a, b), aXkr4 is likely to be stabilized in the presence of Ca²⁺ (Supplementary Fig. 5c, d). Combining the thermostability analysis and Cys and Lys mutant analyses (Supplementary Fig5, Fig. 5i, Fig. 6a-d), it is more likely that Ca²⁺ binds to the pre-activated state of Xkr4 where the XRCC4 fragment binds to Xkr4 dimer.

- In general, the experimental data is intriguing and of very high quality. The weakest piece of evidence in my opinion concerns the blue native gel data shown in Extended Data Fig. 4, which is used to claim that Ca²⁺ binding does not affect dimer stability. While I agree that the data provides qualitative evidence for such behavior, the appearance of a band with faster migrating behavior at 25 °C is apparent in the sample containing EGTA and could indicate the dissociation of the dimer in absence of divalent cations already at room temperature.

Response 3: As described in Response 1 and 2 to reviewer #1's comment, we performed thermostability analysis independently 3 times using both Xkr4 Δ C and aXkr4 dimer. We confirmed that aXkr4, but not Xkr4 Δ C, tends to be stabilized in the presence of Ca²⁺ (Supplementary Fig. 5a-d).

Reviewer #2: *Phospholipid scrambling is an important cell signaling process that remains incompletely understood. Two families of phospholipid scramblases have been identified: TMEM16s and XKRs. Generally, it has been thought that the TMEM16 scramblases are activated by increases in cytosolic Ca and the XKR scramblases are activated by caspase cleavage. This manuscript makes an important contribution by showing that in addition to the requirement for caspase activation, XKR proteins also require extracellular Ca for their scramblase activity. The authors also identify a Ca binding site in the transmembrane region of XKR4, XKR8, and XKR9 that is accessible to the extracellular space. This discovery adds a new dimension to XKR scramblase that may have important implications for understanding the regulation and function of phospholipid scramblases. This is a very nice study with strong data, but there are a few points that need additional development and strengthening.*

We appreciate the reviewer's encouraging comments: "This is a very nice study with strong data, but there are a few points that need additional development and strengthening."

Major Comments:

-Role of intracellular Ca:

The authors suggest that intracellular Ca is not required because ionomycin does not enhance XKR activity. This is not convincing because the question is whether intracellular Ca in the absence of extracellular Ca stimulates XKR4. The best experiment would be to have EGTA extracellular and elevate intracellular Ca with thapsigargin or caged-calcium while measuring cytosolic Ca. This may be a difficult experiment in practice. Another approach would be to buffer intracellular Ca with BAPTA-AM, but the authors should measure intracellular Ca during their manipulations.

Response 1: We appreciate the reviewer's suggestion. To check the contribution of intracellular Ca²⁺, we increased the cytosolic Ca²⁺ using thapsigargin. The result showed that cells treated with 1 μ M thapsigargin had an increased intracellular Ca²⁺ concentration, as detected by Fluo4-AM (shown below and also as Supplementary Fig. 1g in the revised manuscript), however this treatment was not enough to induce PLS (shown below and also as Supplementary Fig. 1h in the revised manuscript). These results are consistent with the A23187 treatment experiment (Fig. 1e), confirming that extracellular, but not intracellular, Ca²⁺ is significant to activate Xkr4.

Supplementary Figure 1: Activation of Xkr4 by XRCC4 mutants. **g** Thapsigargin (Thapsi) treatment on calcium increase. BDKO cells expressing aXkr4 were treated with thapsi at 1 μ M or the same volume of DMSO in the presence of 0.5 mM EGTA. Inner Ca^{2+} concentration was analyzed by FACS using 4 μ M Fluor4-AM. **h** Thapsi treatment on PLS. BDKO cells expressing aXkr4 were treated with 1 μ M thapsi or the same volume of DMSO. Cells were then applied to the PC uptake assay.

-Divalent cation selectivity:

The authors should also test Ba^{2+} and Tb^{3+} . Ba^{2+} is a very important divalent ion to test because it can distinguish between different kinds of Ca^{2+} binding sites. For example, Ba^{2+} binds with different affinity to the two lobes calmodulin, and while Ba^{2+} is more conductive than Ca^{2+} through L-type Ca channels, Ba^{2+} does not support Ca^{2+} - dependent channel inactivation. Tb^{3+} binds very tightly to many Ca^{2+} binding sites in proteins, but in some proteins it acts as an activator and others as an inhibitor. Further, Tb^{3+} fluorescence can be used to study protein conformational changes, so these data could be useful to other investigators for structural studies.

Response 2: We appreciate the reviewer's suggestion. According to the suggestion, we investigated the effects of Ba^{2+} and Tb^{3+} on Xkr4 activity. As a result, we observed weak PLS activity by Tb^{3+} , but not by Ba^{2+} (shown below and also as Fig. 2a in the revised manuscript). The Xkr4 affinity towards Ca^{2+} , Sr^{2+} , Mn^{2+} , and Tb^{3+} was further examined and Tb^{3+} showed much lower affinity to Xkr4 than other ions such as Ca^{2+} and Sr^{2+} (shown below and also as Fig. 2b, c in the revised manuscript).

Fig. 2: Selectivity of divalent cations on Xkr4 activation. **a** Effect of different cations on Xkr4 activity. BDKO cells expressing aXkr4 were incubated with NBD-PC in the presence of 1 mM CaCl_2 , SrCl_2 , MnCl_2 , TbCl_3 , BaCl_2 , FeCl_3 , CoCl_2 , MgCl_2 , NiCl_2 , or ZnCl_2 and analyzed by FACS. **b** Dose-response curves of Xkr4 activity based on PC uptake. BDKO cells expressing aXkr4 were incubated with NBD-PC in the presence of a series of concentrations of CaCl_2 , SrCl_2 , MnCl_2 , or TbCl_3 and analyzed by FACS, followed by quantification. Averages of triplicates were shown with SD. The data for Mn^{2+} at 10 mM and Tb^{3+} at 5 mM and 10 mM were not used for analysis because cells died. **c** Half maximal effective concentration (EC_{50}) of divalent cations on Xkr4 activity. EC_{50} was calculated from the dose-response curve of (b) and shown for each cation. Averages of triplicates were shown with SD.

-I am surprised that Mn^{2+} apparently binds (although at very low affinity). But this raises a possible technical question. Unless great care is taken, even >18M Ω m deionized water can contain ~100 μM Ca^{2+} . And most reagent grade divalent cations contain additional trace Ca contamination. How can the authors rule out Ca contamination of the Mn solution?

Response 3: We obtained $\text{MnCl}_2 \cdot 4\text{H}_2\text{O}$ from Wako (Guaranteed Reagent: purity is 99.0%). The reagent homepage is as described: <https://labchem-wako.fujifilm.com/europe/product/detail/W01W0113-0072.html>. Even if contamination of Ca^{2+} occurs at 1% of Mn^{2+} , Ca^{2+} concentration is 10 μM in 1 mM Mn^{2+} solution, in which concentration Ca^{2+} has no effect on the activity (Fig. 2b). Based on this, we strongly believe that Mn^{2+} has an effect on Xkr4 activity. Following the reviewer's comment, we also examined the PLS activity of aXkr4 in a buffer containing Mn^{2+} or the same volume of distilled water, used for dissolving $\text{MnCl}_2 \cdot 4\text{H}_2\text{O}$. As shown

in the Fig. R1 below, the result supports our initial conclusion that Mn^{2+} can activate aXkr4.

Figure R1: Effect of $MnCl_2$ on Xkr4 activity. BDKO cells expressing aXkr4 were incubated with NBD-PC in the presence or absence of $MnCl_2$ and analyzed by FACS. Buffer, HBSS buffer. DW, distilled water.

-Ca binding site:

-The evidence the authors provide for the Ca^{2+} binding site is strong. But yet, there are certain features of the binding site that seem odd. First, I am surprised that Mn binds even at low affinity. It suggests the binding site is quite flexible since Mn has a much smaller ionic radius than Ca.

Response 4: According to the reviewer's suggestion, we investigated the ionic radius of cations in the literatures (Galperin et al., 2012 *Nucleic Acids Research*; Gupta et al., 2016 *Inorganic Chemistry*; Cheng et al., 2018 *Journal of Membrane Science*) and found that the ionic radius of cations was described as follows: Ba^{2+} (0.136 nm), Sr^{2+} (0.125 nm), Ca^{2+} (0.100 nm), Tb^{3+} (0.092 nm), Mn^{2+} (0.083 nm), and Mg^{2+} (0.072 nm). In this study, we showed that Ca^{2+} , Sr^{2+} , Mn^{2+} , and Tb^{3+} , but not Ba^{2+} and Mg^{2+} , activated Xkr4 with different affinities (Fig. 2a-c), suggesting that appropriate ionic radius (0.083 nm ~ 0.125 nm) is required for cations to bind to the calcium-binding site. These results suggest that the calcium-binding site is flexible to connect TM1 and TM3 in the presence of cations, but Ca^{2+} is the best molecule to connect them to activate Xkr4. This point was described in the main text related to Figure 2 (3rd paragraph of page 4) in the revised manuscript.

-Second, the binding residues are too far apart (6 -9A) in the model to coordinate Ca. This might be expected, because in the absence of Ca, they are repelling one another. But also, this might be a shortcoming of the homology model because the corresponding residues in 7DCE are closer than they are in the XKR4 homology model. Also, the Alpha-Fold model of mXKR4 shows these residues being closer together than the authors' homology model. It might be worthwhile running an MD energy minimization on the XKR4 model (also with Ca added) to see if this improves the binding site configuration. The authors do not provide much detail in their methods how the homology model was constructed or whether there are any probability estimates of confidence of their model. Knowing the architecture of the binding site is important because a 4 – 5 Å rearrangement of the binding site could produce substantial conformational changes elsewhere in the protein that are important

to consider.

Response 5: We understand the reviewer's concern. At the beginning, we compared an AlphaFold model of Xkr4 to a RoseTTAFold model of Xkr4. We concluded that the RoseTTAFold model is more appropriate, based on validation using verify 3D (a tool provided by Institute for Genomics and Proteomics, UCLA) and PROCHECK (a tool provided by European Bioinformatics Institute, EMBL). **This point was described in the main text related to Figure 3(2nd paragraph of page 5) in the revised manuscript.** Although the current model was generated based on the structure of Xkr4 monomer, as described in **Response 1 to reviewer #1's comment**, the Ca²⁺-binding site is coordinated in the pre-active state of Xkr4 such as aXkr4 and Xkr4ΔC binding to XRCC4/C. Therefore, to investigate in detail how the Ca²⁺-binding site is coordinated, a solved structure of aXkr4 or Xkr4ΔC binding to XRCC4/C is required. However, as described in **Response 3 to reviewer #3's comment**, structural analysis of aXkr4 or Xkr4ΔC is currently beyond the scope of this manuscript. At least using the current model of Xkr4 monomer, an MD energy minimization would be difficult to perform and we think that this will be our next challenge for another paper.

-Typically Ca²⁺ is stabilized by 6 - 8 oxygen atoms. The side chains of D123, D127, and E310 may account for 4 contacts, but where are the others? The others could be provided by waters or backbone carbonyls, but it would be nice to have a model of how this binding site would look and how it might change the conformation of the protein.

Response 6: Following the reviewer's comment, we carefully observed the predicted structure of Xkr4 while focusing on the amino acid residues harboring oxygen atoms. We selected T307 and S311 in TM3, and S339 in TM4 as candidate amino acids coordinating the Ca²⁺-binding site. Ala scanning showed that Ala substitution of these amino acids, especially T307A and S339A reduced PLS activity (**shown below and also as Supplementary Fig. 3a-c in the revised manuscript**). Importantly, T307 and S339 are well conserved in mouse and human Xkr8 and Xkr9 (**shown below and also as Supplementary Fig. 3d in the revised manuscript**). These results showed that T307 and S311 on TM3, and S339 on TM4 support D123 and D127 on TM1 and E310 on TM3 forms a calcium-binding site. According to the reviewer's suggestion, we also tried to understand how Ca²⁺ interacts with these amino acid residues using a molecular dynamics (MD) simulation. We performed MD simulation by mimicking the Ca²⁺ binding by essentially fixing the distance of D123, D127 and E310, and found that binding of Ca²⁺ enhanced changes in amino acids movement at local regions synchronously with the contact formation between TM1 and TM3 of aXkr4 (**shown below and also as Supplementary Fig. 7a, b in the revised manuscript**), suggesting that Ca²⁺ binding on TM1 and TM3 induces the conformational change of Xkr4 as it does in the scramblase TMEM16F (Carolina et al., 2019 *eLife*; Simon R et al., 2019 *Nature Communications*; Son C et al., 2020 *Cell Reports*; Arndt et al., 2022 *Nature Communications*). As the published Xkr8 and Xkr9 structure showed a shorter distance between TM1 and TM3 than the modeled Xkr4, cryo-EM or X-ray structural analysis of Xkr4 needs to be performed in the future.

Supplementary Figure 3: Non charged residues present in the calcium-binding site. **a** A Ca^{2+} -binding site of mXkr4. The calcium-binding site-forming amino acids were shown as sticks (oxygen: red). **b** PC uptake assay of BDKO cells expressing aXkr4 with single mutations. BDKO cells expressing each mutant (T307A, S311A, or S339A) were incubated with NBD-PC at a final concentration of 0.125 mM Ca^{2+} or without it, and applied to FACS. **c** Dose-response curves of Xkr4 activity based on NBD-PC uptake. BDKO cells expressing aXkr4 or Ala mutant (T307A, S311A, and S339A), were incubated with NBD-PC at various Ca^{2+} concentrations and analyzed by FACS. Averages of triplicates were shown with SD. The result of aXkr4 in **c** are the same as Fig. 4b. **d** The sequence alignment of human and mouse Xkr4, Xkr8, and Xkr9. The conserved sequence was colored. Candidate calcium-binding sites in TM3 and TM4 were indicated by arrows with amino acid positions according to mXkr4. E310 in the Ca^{2+} -binding site is also indicated by an arrow.

Supplementary Figure 7: Molecular dynamics simulation of Xkr4 with Ca^{2+} . **a, b** Correlation matrix of distance changes of each pair of amino acids. After performing MD simulation and analyzing the trajectories of all amino acids, we calculated how each pair of amino acids changes the distances between the amino acids and the Ca^{2+} position which is postulated to locate at the center of D123, D127, and E310. **a** Correlation matrix of aXkr4 without Ca^{2+} . **b** Correlation matrix of aXkr4 with Ca^{2+} bridge. To mimic Ca^{2+} binding, we applied " Ca^{2+} bridge".

-Cross-linking:

While the cross-linking experiments are suggestive, it is disappointing that the D123C – E310C mutants did not work as expected. Another experiment that might help convince the readers would be to make the D127K or D123K mutants (or D123R or D127R). One would expect these to form salt bridges with E310 if they are close enough to coordinate Ca.

Response 7: According to the reviewer's suggestion, we performed Lys and Arg scanning of amino acid residues coordinating a Ca^{2+} -binding site. Among these mutants, strong PLS activity was observed in aXkr4 E310K without Ca^{2+} , and weak activity was observed in aXkr4 E310R (shown below and also as Fig. 5i in the revised manuscript). These results demonstrated that the E310K forms a salt bridge with D123 and D127 to induce PLS, confirming that Ca^{2+} functions as a molecular glue to connect D123 and D127 on TM1 to E310 on TM3 as shown in Fig. 5a-g.

Fig. 5: Interaction between transmembrane 1 and 3. i Effect of Arg or Lys mutations on Xkr4 activity. BDKO cells expressing aXkr4 with indicated mutations were applied to the PC uptake assay without Ca^{2+} .

-Physiological significance

Because extracellular Ca is physiologically rather constant at ~ 2 mM, changes in Ca occupancy are unlikely to play a physiological role in XKR4 activity unless the affinity of this site is modulated by another mechanism. That is not to say that Ca binding to this site is not important to XKR4 function, but the authors should be more precise how they envision the physiological significance of this binding site.

Response 8: We appreciate the reviewer's comment. Xkr-mediated PLS is known to regulate cell engulfment via phosphatidylserine (PS) exposure. As described in Response 1 to reviewer #3's comment, we examined the effect of extracellular Ca^{2+} on engulfment of Xkr4-expressing cells and found that engulfment of Xkr4-expressing cells by macrophages occurs in an extracellular Ca^{2+} concentration-dependent manner (shown below and also as Supplementary Fig. 1i in the revised manuscript). According to this data, we discussed the physiological significance of Ca^{2+} -binding to Xkr4 in the discussion part (2nd paragraph of page 9) of the revised manuscript as described below.

Compared to ubiquitously-expressed Xkr8, Xkr4 is strongly expressed in the brain¹³ where PS exposure is involved in the removal of unwanted synapses in living neurons³³⁻³⁶. It is well known that extracellular Ca^{2+} is incorporated in activated synapses to regulate neuronal function^{37,38}. Due to the restricted size and limited access to the synaptic cleft, it was predicted that the influx of calcium from the presynaptic and postsynaptic regions during neurotransmission would greatly decrease the calcium concentration within the cleft, potentially reaching as low as 0.3 mM³⁹⁻⁴³. The actual measurement is necessary, but if extracellular calcium is decreased to less than 0.3 mM at the active synapse, it might prevent the engulfment of active synapses by more efficiently suppressing Xkr4 activation.

Supplementary Figure 1: Activation of Xkr4 by XRCC4 mutants. i Engulfment of apoptotic cells by macrophages. Xkr4-expressing PLB cells were stimulated with UV at 2000 J/m² and labeled with pHrodo. Labeled apoptotic cells were incubated with thioglycolate-elicited macrophages in the presence of an indicated concentration of Ca²⁺ for 4 hr and analyzed by FACS.

-Language: The writing could be improved to improve clarity and grammar. A few examples are given below, but these are not comprehensive and the authors should edit their language throughout the manuscript.

Response 9: We appreciate the reviewer’s careful reading of our manuscript. According to the reviewer’s comments, all changes were incorporated in the revised manuscript as below.

The sentence on page 2 “Using three types....” is too convoluted with too many phrases that are hard to parse. We changed the sentence accordingly. Both Xkr4 monomer treated with apoptotic stimuli and Xkr4 dimer stimulated with XRCC4 introduction required extracellular calcium to induce PLS. The constitutively-active mutant of Xkr4 also induced PLS in an extracellular calcium-dependent manner (3rd paragraph of page 2)

The next sentence is incomplete.

Page 3 “We induced cell death with” Would be more clear if it read “in cells expressing either the FL or delta-C form.”

We changed the sentence accordingly. We induced cell death with staurosporine (STS) in PLB cells expressing either the full-length (FL) or the caspase-cleaved form (Δ C) of Xkr4 and analyzed phospholipid scrambling (PLS) activity by assaying incorporation of extracellularly-added fluorescent PC (NBD-PC). (1st paragraph of page 3)

Page 3 “was applied to our PLS activity”. This phrase is nonsense.

We changed this phrase to “was applied to the PLS activity analysis.” (3rd paragraph of page 3)

Page 4 “where negatively charged amino acids were pronounced” is awkwardly phrased.

We changed the phrase to “we aligned mouse and human Xkr4, Xkr8 and Xkr9, focusing on conserved negatively-charged amino acids”. (2nd paragraph of page 5)

Page 4 “to remove the effects of the side chain”. Ala does have a side chain, so this phrasing is odd.

We changed the phrase accordingly. “We generated Ala mutants of each acidic residue”. (3rd paragraph of page 5)

Page 6. “Single Cys mutants in TM1 with or without E310C” seems awkward because E310C is already a single mutant. This could be rephrased.

We changed the phrase accordingly. “Single Cys mutants and double Cys mutants without oxidative reagents were not sufficient to activate aXkr4”. (3rd paragraph of page 6)

Page 6. “lesser” is incorrect

We changed this word to “less”. (3rd paragraph of page 7)

Reviewer #3: Zhang et al. have shown that Xkr4 requires extracellular Ca²⁺ as an additional factor to become a functional lipid scramblase by performing a series of experiments including optical tweezers, structural prediction and biochemical experiments. They also uncovered potential Ca²⁺ binding sites (e.g., D123 and D127 in TM1 and E310 in TM3) in Xkr4. Consistently, its scramblase activity was absent in mutants that neutralize these acidic residues. The present manuscript is very interesting and contains some novel findings, as described above, with supporting biochemical and biophysical data based on their strong expertise.

We greatly appreciate the reviewer’s encouraging comment “The present manuscript is very interesting and contains some novel findings, as described above, with supporting biochemical and biophysical data based on their strong expertise”.

-However, additional important aspects need to be addressed for publication in Nature Communications. The authors need to explain in detail the physiological meaning of extracellular Ca²⁺-mediated Xkr4 functions. For example, it is essential to show whether extracellular Ca²⁺ acts as a molecular switch for the physiological and/or pathophysiological functions of Xkr4.

Response 1: We appreciate the reviewer’s comment. The physiological function of Xkr family members has been known to regulate cell engulfment by phagocytes through exposure of phosphatidylserine (PS) (Suzuki et al, 2013 *Science*; Suzuki et al., 2014 *J Biol Chem*; Kawano and Nagata, 2018 *PNAS*). Therefore, we examined whether extracellular Ca²⁺ acts as a molecular switch for engulfment of Xkr4-expressing cells. Ultraviolet light was used to induce apoptosis in Xkr4-expressing cells, and pHrodo-labeled apoptotic cells were incubated with macrophages in the presence or absence of extracellular Ca²⁺. As shown in Supplementary Fig.1i (also see Response 8 to reviewer #2’s comment), macrophages engulfed the Xkr4-expressing apoptotic cells in an extracellular Ca²⁺ concentration-dependent manner, suggesting that extracellular Ca²⁺ functions as a molecular switch for engulfment. This point was described in 1st paragraph of page 4 in the revised manuscript. The effect of extracellular Ca²⁺ on engulfment in the brain (where Xkr4 is strongly expressed) was described in Response

8 to reviewer #2's comment and discussion part of the revised manuscript. (2nd paragraph of page 9)

-It is also important to prove the structural changes via coordination of Ca²⁺ in the TM1 and TM3 of Xkr4 by performing intensive structural analysis. Therefore, this manuscript is not suitable for publication in this esteemed journal in its present form.

Response 2: As the reviewer claimed, we do believe that structural analysis strongly supports our finding. Indeed, we have been attempting a structural analysis of active Xkr4 for the past 3.5 years. Recently, monomeric structures of Xkr8 and Xkr9 have been solved by cryo-EM analysis (Monique et al., 2021 *eLife*; Sakuragi et al., 2021 *Nat Struct Mol Biol*), but structures of active Xkr8 and Xkr9 have not been solved yet and are next big challenge in this field. We have tried to express and purify Xkr4 dimer from FreeStyle 293F cells using the baculovirus expression system, and Xkr4 dimer was successfully purified at a low concentration as described below (Fig. R2a, b). However, at a high concentration (>1.0 mg/ml), purified Xkr4 dimer tend to become unstable especially when applied to cryo-EM analysis as described in below (Fig. R2c). We attempted to improve the stability of Xkr4 by inserting mutations, truncating some regions, or inserting the protein into nanodiscs, but we were not successful at performing cryo-EM analysis. Therefore, we believe that structural analysis of Xkr4 dimer is technically beyond the scope of current manuscript and it needs to be reported in another paper.

Figure R2: Cryo-EM structure analysis of human Xkr4ΔC (hXkr4 ΔC). **a** Size exclusion chromatography (SEC) of hXkr4ΔC. The hXkr4ΔC, purified by affinity purification using nickel column, was applied to SEC and the dimer peak was collected at 39 to 44 min (left, black). Collected dimer was applied to SEC (left, green) and BN-PAGE, followed by detection with anti-V5 antibody (right). **b** Negative staining of purified Xkr4 dimer. Negative staining reflects the homogeneity of purified Xkr4 dimer. Purified Xkr4 was diluted to 30 μg/ml (left) and 3 μg/ml (right). **c** A representative two-dimensional image. A high-quality two-dimensional image was highlighted by a red square.

Instead, in the revised manuscript, we intensively analyzed the effect of Ca^{2+} on TM1 and TM3 contact formation. As described in Response 1 of reviewer #1's comment, E310K mutation activated aXkr4, but not Xkr4FL and Xkr4ΔC, without extracellular Ca^{2+} (Fig. 5i, Supplementary Fig. 6b), suggesting that the pre-activated form of Xkr4 facilitates PLS with TM1 and TM3 contact. This conclusion was supported by Response 2 and 3 of reviewer #1's comment: Ca^{2+} stabilizes the conformation of aXkr4, but not Xkr4ΔC (Supplementary Fig. 5a-d). In addition to these data, artificial disulfide bond formation between TM1 and TM3 (Fig. 5a-g) strongly demonstrated that Ca^{2+} regulates TM1 and TM3 contact formation to induce PLS. To understand how Ca^{2+} activates Xkr4 (as described in Response 6 of reviewer #2's comment), we performed a molecular dynamics (MD) simulation to mimic a Ca^{2+} binding by effectively fixing the distance among Ca^{2+} binding sites (D123, D127, and E310). As a result, we found that binding of Ca^{2+} enhanced changes in amino acids movement at local regions synchronously with the contact formation between TM1 and TM3 of aXkr4, suggesting that Ca^{2+} binding on TM1 and TM3 induces the conformational change of Xkr4 (Supplementary Fig 7a, b).

-Although the authors conclude that Xkr4 activity regulates membrane tension, the molecular mechanism is not fully addressed in the present manuscript. To strengthen the significance of this manuscript, it is important to show whether Xkr4-mediated phospholipid scrambling directly regulates intrinsic membrane properties or whether it indirectly affects membrane tension via the rearrangement of the intracellular cytoskeleton.

Response 3: It has been known that membrane tension is altered as a consequence of PLS (Shiomi et al., 2021 *Cell Reports*; Tsuchiya et al., 2018 *Nature Communications*). In the present manuscript, we analyzed the membrane tension to see the effect of Ca^{2+} -mediated PLS from the different aspect, but not to understand mechanisms of alteration in membrane tension. Following the reviewer's comment, we also visualized the actin cytoskeleton using Alexa Fluor 488 phalloidin in cells with or without 2 mM Ca^{2+} . As shown in Fig. R3 below, F-actin showed no obvious changes with or without Ca^{2+} , suggesting that Xkr4 regulates membrane tension via PLS without affecting the cytoskeleton in our system. However, the mechanism of alteration of the membrane tension is not a main focus in this manuscript. Therefore, we prefer not to describe it in the revised manuscript. However, we will describe about this in the manuscript with data if the reviewer recommends to add.

Figure R3: F-actin in BDKO cells. BDKO cells expressing αXkr4 were stained by Alexa Fluor 488 phalloidin (green) to see F-actin and by DAPI (blue) to see nucleus in the presence (right) or absence (left) of Ca^{2+} . Scale bar, 20 μm .

-In optical tweezer experiments, why did the authors use an external solution containing 5 mM Ca^{2+} ? To evaluate physiological membrane properties, the concentration of Ca^{2+} should be approximately 2 mM.

Response 4: According to the reviewer's comment, we examined the effect of 2 mM Ca^{2+} on membrane tension, and found that the effect of 2 mM Ca^{2+} on membrane tension is comparable to 5 mM Ca^{2+} (shown below and also as Fig. 1g, h in the revised manuscript).

Fig. 1: Requirement of extracellular calcium for Xkr4-mediated PLS. **g** Force-extension curve used to obtain membrane tension for BDKO cells expressing aXkr4 in the presence of 2 mM (pink) or 5 mM Ca²⁺ (green), or absence of it (black). Solid lines are the linear fit used to obtain membrane tension. **h** Membrane tension of BDKO cells expressing aXkr4. Box-plots of membrane tension of aXkr4 in the presence of 2 mM (pink) or 5 mM Ca²⁺ (green), or absence of it (black). ns, not significant. **, $p < 0.01$.

-To evaluate the role of external Ca²⁺ in Xkr4 function as a phospholipid scramblase in detail, the authors are encouraged to measure Xkr4-mediated translocation of other phospholipids, such as phosphatidylserine and phosphatidylethanolamine.

Response 5: Following the reviewer's comment, we investigated the PLS activity of Xkr4 for other phospholipids such as PS and SM. To observe PS exposure, we cannot use Annexin V that is often used to detect PS because it requires Ca²⁺ to bind to PS. Therefore, we used MFG-E8 that binds to PS without Ca²⁺. MFG-E8 was fused with monomeric EGFP, expressed in CHO cells, and purified from culture supernatant. Using purified MFG-E8-EGFP, we showed that aXkr4 exposes PS in an extracellular calcium-dependent manner (shown below and also as Supplementary Fig. 1f in the revised manuscript). We also performed an NBD-SM uptake analysis and confirmed that aXkr4 also incorporates SM (shown below and also as Supplementary Fig. 1e in the revised manuscript).

Supplementary Figure 1: Activation of Xkr4 by XRCC4 mutants. **e** Sphingomyelin (SM) uptake assay of BDKO cells expressing aXkr4. Cells were incubated with NBD-SM in the presence or absence of 1 mM Ca²⁺ for 40 min. **f** PS exposure assay of BDKO cells expressing aXkr4. Cells were incubated with EGFP-fused MFG-E8 in the presence or absence of 1 mM Ca²⁺ for 40 min.

Although structural analysis was not performed in this study, we believe that our revised manuscript is suitable for publication in Nature Communications as noted by reviewer #1 *“I am convinced that the manuscript will be of large importance for the field and thus consider it as strong candidate for publication in Nature Communications. I do not think that the study requires more data and only have few remarks that might improve the clarity of the work”*. Reviewer #2 also noted that *“This is a very nice study with strong data, but there are a few points that need additional development and strengthening.”* In the revised manuscript, additional data were successfully incorporated to strengthen our finding.

Once again, we wish to express our sincere appreciation to the reviewers for their constructive comments.

REVIEWERS' COMMENTS

Reviewer #1 (Remarks to the Author):

I think that the authors have responded to the comments of the reviewers in a satisfactory manner. The manuscript has further improved and should in my opinion be accepted for publication in its current form.

Reviewer #2 (Remarks to the Author):

The authors have thoroughly addressed all of my concerns. This is a very nice manuscript and makes an important contribution to the field. I apologise for my delay in reviewing the manuscript - some health and family issues interfered.

Reviewer #3 (Remarks to the Author):

The authors adequately addressed my comments on the initial submission. I can agree that some points raised might go beyond the scope of this manuscript. In my opinion, the revised manuscript is now suitable for publication.